# Detection of viral RNAs at ambient temperature via reporter proteins produced through the target-splinted ligation of DNA probes

Elizabeth A. Phillips [1,9], Adam D. Silverman [1,9], Aric Joneja [1,9] ✉, Michael Liu[1], Carl Brown[1,2], Paul Carlson[1], Christine Coticchia[1], Kristen Shytle [1], Alex Larsen[1], Nadish Goyal[1], Vincent Cai[1], Jason Huang[1], Jennifer E. Hickey[1], Emily Ryan[1], Joycelynn Acheampong[1], Pradeep Ramesh[1], James J. Collins [2,3,4,5,6,7,8] & William J. Blake[1,2]

Nucleic acid assays are not typically deployable in point-of-care settings because they require costly and sophisticated equipment for the control of the reaction temperature and for the detection of the signal. Here we report an instrument-free assay for the accurate and multiplexed detection of nucleic acids at ambient temperature. The assay, which we named INSPECTR (for internal splint-pairing expression-cassette translation reaction), leverages the target-specific splinted ligation of DNA probes to generate expression cassettes that can be flexibly designed for the cell-free synthesis of reporter proteins, with enzymatic reporters allowing for a linear detection range spanning four orders of magnitude and peptide reporters (which can be mapped to unique targets) enabling highly multiplexed visual detection. We used INSPECTR to detect a panel of five respiratory viral targets in a single reaction via a lateral-flow readout and ~4,000 copies of viral RNA via additional ambient-temperature rolling circle amplification of the expression cassette. Leveraging synthetic biology to simplify workflows for nucleic acid diagnostics may facilitate their broader applicability at the point of care.

The ongoing COVID-19 pandemic underscores the urgent requirement for new and innovative diagnostic technologies to enable low-cost detection of pathogens at the point of care and point of need[1]. Rapid antigen tests dominate on-demand testing around the world due to

their ease of use, low cost and room-temperature storage[2]. However, the identification of antigen-specific antibodies required for the manufacturing of such tests is challenging and time-intensive, limiting the capability of antigen tests to multiplex or distinguish pathogen strains

[1]Sherlock Biosciences, Watertown, MA, USA. [2]Wyss Institute for Biologically Inspired Engineering, Harvard University, Boston, MA, USA. [3]Institute for Medical Engineering and Science, Department of Biological Engineering, Massachusetts Institute of Technology, Cambridge, MA, USA. [4]Infectious Disease and Microbiome Program, Broad Institute of MIT and Harvard, Cambridge, MA, USA. [5]Abdul Latif Jameel Clinic for Machine Learning in Health, Massachusetts Institute of Technology, Cambridge, MA, USA. [6]College of Arts and Sciences, Harvard University, Cambridge, MA, USA. [7]Synthetic Biology Center, Massachusetts Institute of Technology, Cambridge, MA, USA. [8]Harvard-MIT Program in Health Sciences and Technology, Massachusetts Institute of Technology, Cambridge, MA, USA. [9]These authors contributed equally: Elizabeth A. Phillips, Adam D. Silverman, Aric Joneja. ✉e-mail: ajoneja@sherlock.bio

and variants. Molecular assays that detect viral or bacterial nucleic acids are preferable to rapid antigen tests for their faster development cycles, better discrimination of variants, higher sensitivities and superior limits of detection (LoD)[3]. Still, the gold standard for such assays, quantitative polymerase chain reaction (qPCR), requires sophisticated instrumentation to control temperature cycling and detect amplification of specific target sequences, and is challenging to deploy at the point of need[4]. An instrument-free assay and detection method could dramatically improve the accessibility of sensitive molecular tests to consumers, particularly those who lack access to centralized testing labs[5].

The precise cyclical heating requirements of qPCR have been addressed by isothermal nucleic acid amplification strategies, such as loop-mediated amplification (LAMP)[6], recombinase polymerase amplification (RPA)[7,8] and nicking enzyme amplification reaction (NEAR)[9]. Many of these strategies couple amplification to a detection step employing fluorescent probes (such as RPA) or molecular beacons (such as NEAR) to provide an additional layer of specificity. Other methods, such as SHERLOCK or DETECTR, exploit CRISPR technology to detect amplification via the release of a quenched fluorophore using the collateral activity of the programmable nucleases Cas12 or Cas13 (refs. [10–17]). However, these isothermal strategies still require instruments: portable heaters to achieve and maintain the optimal temperature for enzyme activity during the exponential amplification and photometers to measure fluorescence. Moreover, such isothermal amplification techniques have limited capability for multiplexing due to challenges in primer design and the overlap of fluorescent reporters' emission spectra.

By contrast, synthetic biology strategies enable the detection of nucleic acids with more flexible protein-based readouts. Gene expression can be gated using a sequence-specific riboregulator, such as a toehold switch, such that only the target (or 'trigger') RNA sequence transduces a signal to activate production of a protein reporter[18]. Since these reporters can be fluorescent, colorimetric or luminescent, a variety of detection modes are possible. Toehold switches, in particular, have been deployed for in vitro molecular diagnostics in low-resource settings[19]. The pathogen-derived target sequences are added directly to a freeze-dried cell-free expression (CFE) reaction[20], often after pre-amplification, enabling shelf-stable and sensitive detection of Ebola[20], Zika[21], norovirus[22], *C. difficile*[23], HIV[24] and SARS-CoV-2 (ref. [25]) sequences. Although these cell-free expression reactions can be run at close-to-ambient conditions, the requisite pre-amplification often means that equipment is still required.

Here we report INSPECTR, a modular, ambient-temperature strategy for sensitive, specific, multiplexable and quantitative nucleic acid detection. Similar to strategies used for transcriptomic profiling (such as cRASL-seq[26], TempO-Seq[27] or SplintQuant[28]) or aptamer-based diagnostics (such as SENSR[29] or SNAILS[30]), RNA is detected by the splint-ligation of designed DNA probes. However, in INSPECTR, ligation directly generates an expression cassette programmed for the cell-free synthesis of a reporter protein. This strategy allows the translation of any polypeptide to be triggered by the presence of a specific unrelated nucleic acid sequence, which in turn enables highly multiplexed assays using a lateral-flow readout. INSPECTR avoids amplification of the target; instead, detection is achieved through cell-free expression of enzymes or peptides and, if desired, ambient temperature amplification of the ligated probe. We implement our workflow in the context of detecting viral genomic RNA and demonstrate both sensitive (limit of detection of approximately 4,000 copies) and specific multiplexable detection of a panel (*N* = 5) of respiratory viral target sequences, all at ambient temperature. We anticipate that the modularity and versatility of the INSPECTR platform will contribute an important tool to the arsenal of synthetic biology-based strategies for molecular diagnostics and spur further innovation and use of cell-free expression systems for ensuring global health.

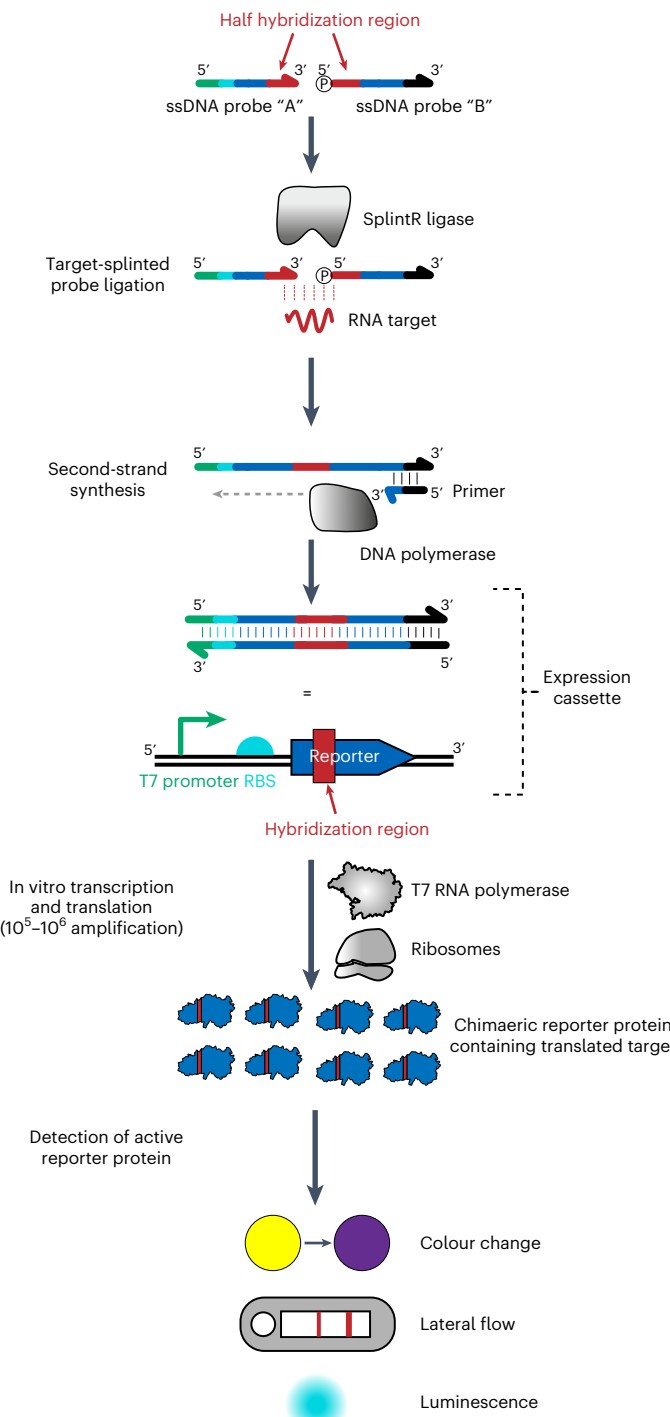

**Fig. 1 | INSPECTR process workflow.** Single-stranded DNA probes, designed to have homology against a particular target RNA, are ligated only when the RNA splints a ternary molecular junction. The ligated probes serve as a template for double-stranding by a DNA polymerase. The linear double-stranded product encodes a complete reporter expression cassette, including a T7 promoter, a ribosome-binding site (RBS), a reporter protein and a T7 terminator. When added to a cell-free expression system, transcription and translation generate a modular reporter output that can be measured electronically or visually.

## Results

### INSPECTR workflow

An outline of the INSPECTR workflow is diagrammed in Fig. 1. In the first step, single-stranded DNA (ssDNA) probes hybridize to an RNA target that acts as a splint for ligation by *Chlorella* virus DNA ligase

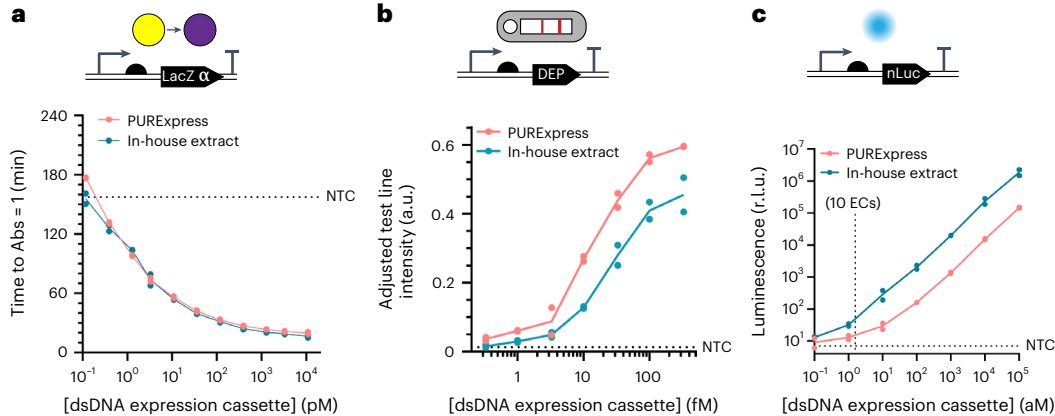

**Fig. 2 | Low-copy detection of reporter proteins in cell-free expression systems. a–c**, After physiochemical optimization, mock 'pre-ligated' dsDNA expression cassettes could be detected down to (**a**) ~1 pM (colorimetric reporter, where easy visual detection was assumed for absorbance ≥1.0), (**b**) ~1 fM (peptide reporter, lateral-flow assay) and (**c**) ~1 aM or 10 expression cassettes (luminescent reporter, plate reader) after 2 h of expression at 22 °C in both a purified system

(PURExpress in vitro protein synthesis kit) and crude *E. coli* lysate. Experimental details are provided in Methods. Plotted data represent technical replicates and their mean at the indicated concentration of expression cassette (EC). The corresponding signal or time-to-result for the no-template control (NTC) is plotted with a horizontal dotted line.

(SplintR), similar to the padlock probe methodology used for micro-RNA detection[31]. Next, a DNA polymerase synthesizes the complementary strand of the ligated ssDNA probe to generate a double-stranded DNA (dsDNA) expression cassette. This expression cassette encodes a reporter protein under control of transcription by T7 RNA polymerase. When added to a cell-free expression system containing ribosomes, T7 RNA polymerase, translation factors, amino acids, nucleotides and additional cofactors[32–34], the expression cassette yields a reporter protein that can be detected visually or electronically. The entire INSPECTR workflow occurs at ambient temperature and does not require nucleic acid pre-amplification. Moreover, this design strategy decouples input sequence from expression output, creating the flexibility to pair any target nucleic acid with a wide array of reporter proteins.

We began our design process by evaluating readily detectable reporter protein outputs for the downstream cell-free expression (CFE) reaction. Conventionally, cell-free expression and related biosensors rely on high concentrations of DNA templates (10 nM or higher), which are regulated at the transcriptional or translational level[35]. Using the INSPECTR mechanism, each target nucleic acid will only 'activate' one pair of ssDNA probes to form a single dsDNA expression cassette; we therefore expected that INSPECTR would need to generate a detectable reporter from very small numbers of expression cassettes. Unlike previous efforts in CFE optimization in which DNA template concentration is not a limiting factor, we sought to optimize the reporter signal per expression cassette. We designed linear dsDNA expression cassettes for three putative reporter protein outputs: beta-galactosidase (LacZ), which can be detected visually and has been extensively used in cell-free biosensors[22,36]; a 3x-FLAG- and Twin-Strep-tag-containing peptide, which can be detected using a lateral-flow assay; and nanoluciferase (nLuc), which catalyses a light-emitting reaction detectable on a plate reader. We then estimated the feasible limit of detection for each reporter.

By expressing the small 45-amino acid alpha-fragment of LacZ and adding the purified omega-fragment complement, we could detect a colour change when a DNA expression cassette was supplied at concentrations of around 1 pM (Fig. 2a). However, since INSPECTR does not amplify the target sequence, this level of sensitivity would require at least 60 million target copies per reaction, which is outside the range of clinical relevance.

The peptide and nLuc reporter outputs were far more sensitive. Using both a commercially available reconstituted transcription-translation reaction with purified enzymes (PURExpress[37]) and an in-house generated extract[38], we detected as low as 1 fM of the dsDNA

linear expression cassette encoding a reporter peptide and around 1 aM of the cassette encoding nLuc (<10 expression cassettes per reaction) (Fig. 2b,c). Comparing the linear dose-response curves of dsDNA template and purified nLuc, we estimate that transcription and translation provide amplification by a factor of $10^5$–$10^6$. Consistent with previous models, at low template concentrations (<1 pM), protein yield scales linearly with the amount of template provided[39]. Since each reporter protein would have different properties as an output of the INSPECTR reaction, we evaluated the enzymes and peptides separately.

## RNA detection using a luminescent INSPECTR output

The highly sensitive and linear detection of mock pre-ligated dsDNA in Fig. 2c suggested that nLuc could be a useful reporter output for INSPECTR due to its small size and high luminosity[40]. We therefore designed luminescent probes to detect the SARS-CoV-2 RNA genome. Because the hybridization sequence in an INSPECTR probe is necessarily present in the final expression cassette (Fig. 1), we first searched for permissive insertion sites in the coding sequence (CDS) of nLuc, which would not impact the expression or function of the translated enzyme. We tiled four 30 nt DNA sequences, complementary to conserved regions of interest (ROIs) in the SARS-CoV-2 genome, at diverse structural contexts in CDS of the 171-aa nLuc protein. For the design of the target regions, we aimed to maximize conservation in viral genomes above any sequence preferences, noting previous studies that indicate SplintR ligates RNA well as long as the overhangs are longer than 4–6 nt (refs. 41,42). We kept the junction roughly symmetric and tried to ensure that the 5′ phosphate donor nucleotide was not a dG or dC, in line with the enzyme's sequence preference. When expressed at low-copy (2 fM of expression cassette) in CFE reactions, we found that approximately half of these insertions abrogated either protein synthesis or function relative to the wild type (WT, Fig. 3a). For a subset of the insertion variants that did not impact function, we constructed and tested split ssDNA probes for detection of the targeted RNA sequences and identified one probe set (split after the eleventh residue in nLuc) with high signal only in the presence of the target.

Next, we tested this probe's function in a two-step, 50 min workflow where the ligated probe was double-stranded during the expression reaction (Fig. 3b). In this ambient temperature assay, the nLuc probe detected 1 fM of short synthetic target RNA and 10 fM of extracted SARS-CoV-2 RNA, corresponding to ~1,000 genome copies in the ligation reaction. As before, the response was highly linear across a concentration range spanning four orders of magnitude. Since

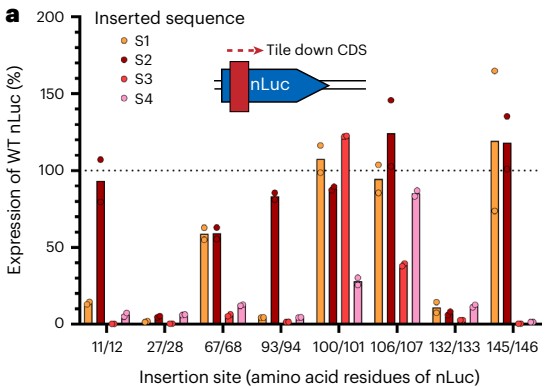
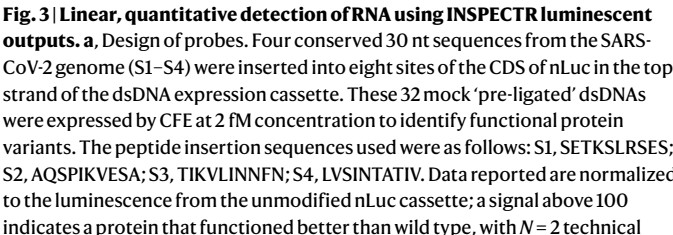
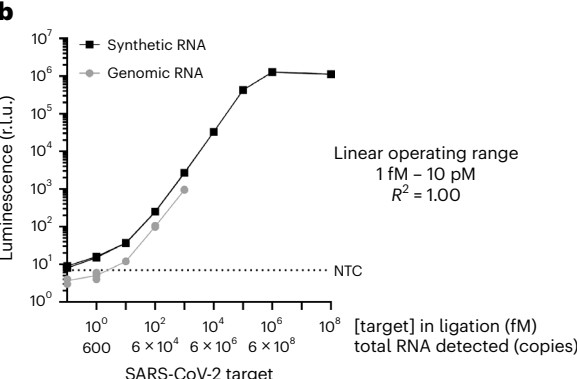

**Fig. 3 | Linear, quantitative detection of RNA using INSPECTR luminescent outputs. a**, Design of probes. Four conserved 30 nt sequences from the SARS-CoV-2 genome (S1–S4) were inserted into eight sites of the CDS of nLuc in the top strand of the dsDNA expression cassette. These 32 mock 'pre-ligated' dsDNAs were expressed by CFE at 2 fM concentration to identify functional protein variants. The peptide insertion sequences used were as follows: S1, SETKSLRSES; S2, AQSPIKVESA; S3, TIKVLINNFN; S4, LVSINTATIV. Data reported are normalized to the luminescence from the unmodified nLuc cassette; a signal above 100 indicates a protein that functioned better than wild type, with $N$ = 2 technical

replicates. Error bars indicate standard deviation of the mean. **b**, Luminescent INSPECTR assay. The probe set corresponding to insertion sequence S2 at residues 11/12 was ligated on either synthetic or genomic SARS-CoV-2 RNA sequence in a SplintR ligase reaction at 22 °C. After 30 min, the reaction was diluted 1:2 into a PURExpress CFE reaction and luminescence was monitored continuously for 20 min. Plotted data represent the endpoint luminescent signal from three independent biological replicates from separate ligations. (Individual replicates are plotted but not visible due to marker size.) $R^2$ was calculated after log–log transformation of the data in GraphPad Prism (v9.4.0).

there is no exponential amplification of the target in this workflow, the RNA target can be effectively quantified directly from a linear calibration curve. We measure a ~1,000-fold loss of sensitivity from the ligation-expression workflow, when compared with the 'mock pre-ligated' dsDNA experiments in Fig. 2c. Previous reports suggest SplintR ligation never runs to completion, even at high temperatures and at high concentrations of enzyme relative to the DNA substrates[43]. This results in a loss of sensitivity that is compounded by inefficiencies in template double-stranding, poisoning of the expression reaction by the ligation buffer and dilution between reaction steps. Further optimizations on ssDNA probe architecture or ligase engineering will probably be necessary to further improve the sensitivity of this two-pot assay.

### Multiplexed RNA detection using dual-epitope peptide reporters

While a luminescent reporter enabled quantitative RNA detection at ambient temperature, we aimed to develop INSPECTR into a truly instrument-free assay that leverages the ease of use of rapid antigen tests. First, to shorten the typical process redesign cycle for a target and improve the 'hit rate' of functional probes described in Fig. 3a, we developed a reporter detection strategy that would be universal to any RNA target input. We selected a sandwich immunoassay strategy to leverage the specificity enabled by two analyte binding events. We hypothesized that the two binding events would reduce the background detection of spurious outputs caused by expression of a partially formed expression cassette. We then sought epitopes that could flank the peptide reporter on the N and C termini and be detected by a universal lateral-flow immunoassay. To select high-affinity antibody and epitope pairs that enable lateral-flow detection, we generated screening libraries of peptides flanked by epitope candidates, where each library was sampled from 20 (ref. 12) possible peptide reporter structures. The peptide library was synthesized by cell-free expression to preserve the structure of INSPECTR protein outputs and to eliminate any antibody candidates that cross-react with CFES components. Monoclonal antibodies were screened against the epitope candidates in a lateral-flow immunoassay format as both capture (striped on nitrocellulose membrane) and detection (actively conjugated to gold nanoparticles) antibodies (Fig. 4a, left).

Functional antibody and epitope pairs were selected for further lateral-flow development and optimization to improve sensitivity.

We validated expression of epitope candidates in both PURExpress and a crude extract (NEBExpress), but ultimately shifted to NEBExpress so that we could include the His epitope in our reporter library; the enzymes of PURExpress are His-tagged for purification and outcompeted the capture of dual-epitope peptides using a His-tag. Consistent with previously reported uses of epitope tags, we found that some epitopes functioned better on either the N or C terminus of the reporter, probably due to improved cell-free translation. We also found that the inclusion of the 12-amino acid translation of the RNA target sequence, serving as a linker between the two epitopes of the peptide reporter, reduced the risk of antibody steric hindrance. Further, by using multimeric epitopes, we improved detection sensitivity 100-fold (for example, 3x-FLAG and Twin-Strep-tag) (Supplementary Fig. 2 (monomeric vs multimeric detection)). Through our selection strategy, we identified antibodies that robustly captured and detected dual-epitope peptide reporters despite variations in internal sequences. To validate the universal detection of a 3x-FLAG (N-terminus) and Twin-Strep-tag (C-terminus) dual-epitope reporter, we designed 187 expression cassettes each encoding for a 3x-FLAG and Twin-Strep-tag reporter but with unique, 36-base-pair internal sequences representing ROIs tiling the SARS-CoV-2 genome. After a 2 h ambient-temperature expression reaction, the output of each expression cassette was mixed with a running buffer and nanoparticles conjugated to anti-*Strep*-tag II antibody. The dual-epitope peptide reporter then wicked through a lateral-flow half-strip assay with an anti-FLAG antibody test line. All 187 peptide reporters that were generated from 10 pM of expression cassette developed a visually detectable line on the universal lateral-flow strip (Fig. 4a, right), indicating the target-agnostic universality of the dual-epitope reporter approach.

Given cell-free systems' inherent ability to simultaneously express multiple outputs[44,45], INSPECTR enables multiplexing by addressing each reporter output to a unique RNA input. First, we sought to resolve simultaneously expressed peptide reporters on a single lateral-flow strip. To minimize the risk of off-target signals during multiplexing, previously screened antibodies were validated for low cross-reactivity. Antibodies that retained epitope-specific activity in a lateral-flow format were then selected for development into a spatially multiplexed test strip (Fig. 4b). To account for the variability in the antibodies' affinity to their respective epitopes, we empirically optimized the

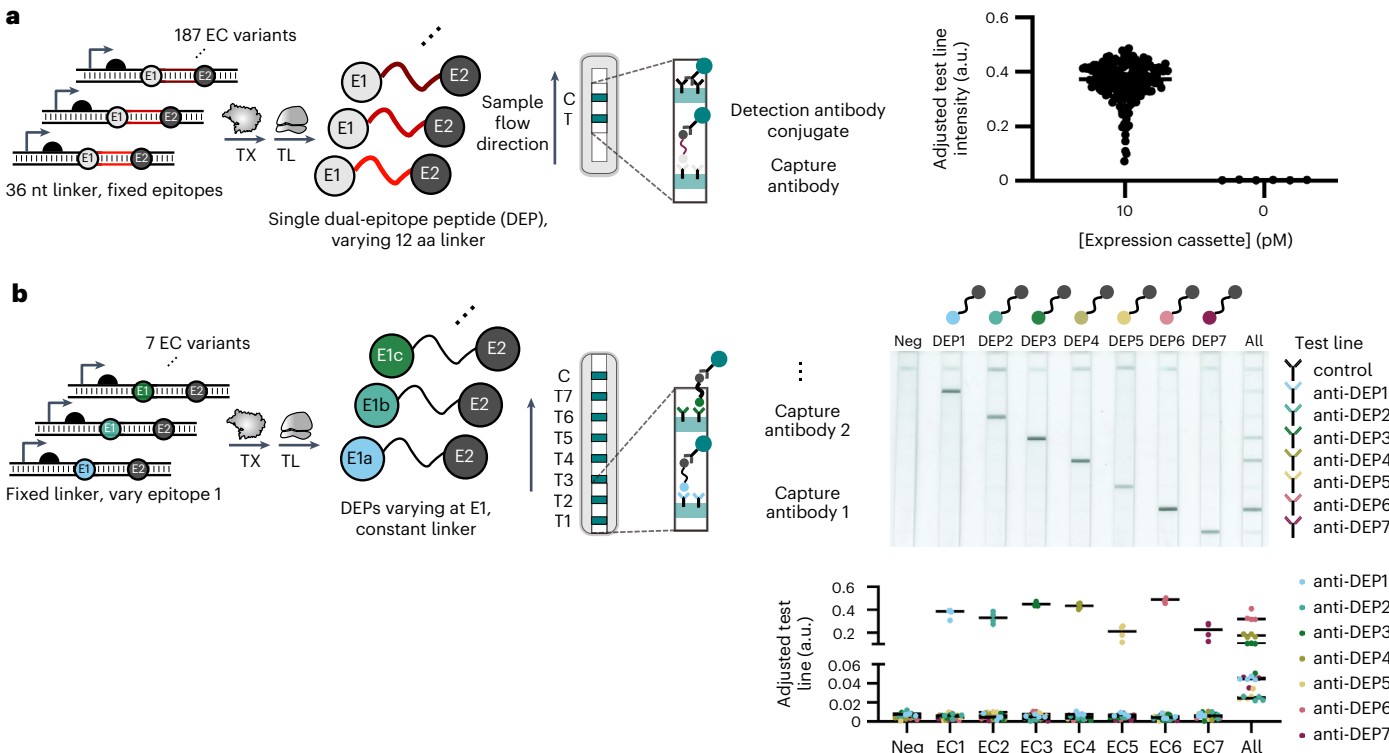

**Fig. 4 | Dual-epitope peptide design enables specific, programmable lateral-flow immunoassay detection.** Expression cassettes encode reporter peptides with 36-base-pair internal sequences flanked by distinct N- and C-terminal epitopes that enable detection by a sandwich lateral-flow immunoassay. The intensity of developed test line(s) can be visually assessed or scanned and analysed by ImageJ to subtract out test membrane background. **a**, Left: detection of 187 unique dual-epitope peptides with 3x-FLAG (E1, N-terminal) and Twin-Strep-tag (E2, C-terminal) epitopes by universal lateral-flow immunoassay. Each peptide's internal sequence represents a 12-amino acid sequence translated from 36-base-pair sequences tiled from the SARS-CoV-2 genome. Right: data points

represent background-subtracted test line intensity of 187 peptides' detection with technical replicates. **b**, Left: specific detection of 7 unique dual-epitope peptides with unique N-terminal epitope, SARS-CoV-2 internal sequence and Twin-Strep-tag (E2, C-terminal) epitope by lateral-flow immunoassay. Expression cassette of each peptide added to cell-free expression system generates a dual-epitope peptide that is specifically captured by spatially multiplexed test line antibodies. Top right: scanned images of representative test strips. Bottom right: plotted data representing background-subtracted test line intensity of two technical replicates of independent biological triplicates and their mean.

antibodies' striping order and buffer composition. Antibodies found to have the highest sensitivity in a 1-plex format were generally striped closer to the control line to maximize the detection conjugate available for weaker-binding antibodies. Resuspending test line antibodies into a striping buffer containing sugars also improved wettability at the test lines and reduced non-specific binding. To demonstrate the optimized test strips' specificity to reporters, we designed seven expression cassettes, each encoding a unique N-terminal epitope but a common 12-amino-acid internal sequence and C-terminal epitope (Twin-Strep-tag). A single detection antibody conjugate targeted the Twin-Strep-tag tag of every reporter peptide, which could then be specifically captured on its respective test line. We generated a set of seven orthogonal peptide outputs that could be captured on a single strip (Fig. 4b). We noticed that when all reporters were co-expressed in a single reaction, the test signal was reduced, possibly due to competition for transcription-translation resources or for the detection conjugate. In the future, this competition could be mitigated by redesigning the reporters to have unique C-terminal epitopes and using multiple detection conjugates.

To demonstrate the utility of this multiplexed INSPECTR assay, we screened INSPECTR peptide probes targeting three ROIs each from five respiratory pathogens: the SARS-CoV-2 Wuhan and Delta variants, influenza A virus (IAV), influenza B virus (IBV) and respiratory syncytial virus (RSV). For each ROI, we designed probes to encode each of the reporter peptides previously detected by the universal and multiplexed test strips. To reduce the possibility of probe self-splinting, which could

lead to ligase-dependent background[46], we included two consecutive ligation junctions in each probe, spanned by a short 10 nt gap-filling oligonucleotide (GFO) (Fig. 5a). This strategy reduces the assay background because transcription of any improperly ligated expression cassettes would generate an mRNA with a −1 frameshift mutation in the second epitope, producing a peptide that cannot be detected by lateral-flow assay. Using a model probe and quantitative readout by PCR and cell-free expression, we determined that adding this second ligation junction resulted in a small (~50%) reduction in ligation efficiency and ON signal, but greatly reduced probe 'leak' (ligase-independent expression) (Extended Data Fig. 1).

Each probe was then evaluated for its transduction of a synthetic target RNA into a reporter peptide, as detected on a single-plex test strip. Using our new design, only three of the 120 probes tested generated background lateral-flow signal in the absence of RNA target. The probes that generated reporters with OLLAS or Myc epitopes yielded lower test signal than other reporters due to lower test line antibody capture affinity and were eliminated from further testing. To evaluate probe multiplexing, we pooled a subset of the remaining probes into combinations of five probe sets each that would encode a unique reporter for each pathogen (Fig. 5a). The addition of each individual synthetic RNA target into an optimized pool of all probes yielded a unique peptide reporter that was specifically captured on distinct test lines of a multiplexed test strip. Similarly, the addition of all five RNA targets into the pool also generated peptides that developed into visually detectable test lines on the test strip (Fig. 5b). Ultimately, we

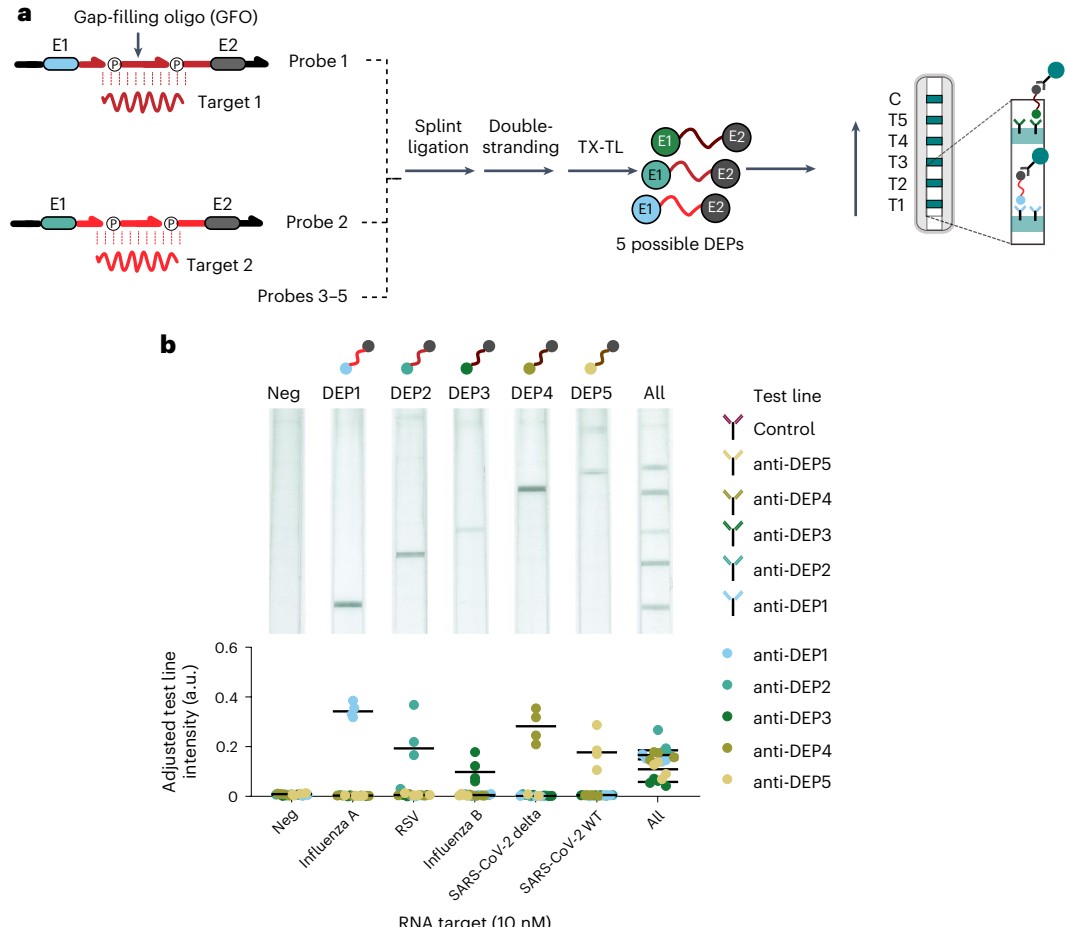

**Fig. 5 | Multiplexed detection of synthetic RNA targets representing five respiratory pathogens. a**, Generation of five unique reporter peptides transduced by RNA target-templated ligation of DNA probes. Peptides are specifically captured by spatially multiplexed test line antibodies. **b**, Specific detection of peptides transduced by RNA target recognition. A single RNA target introduced to a cocktail of DNA probes yields a peptide that is specifically captured by pre-designated test line. Addition of all five RNA targets to a cocktail of DNA probes enables co-expression of each peptide reporter and subsequent capture on pre-designated test lines. Top: scanned images of representative test strips. Bottom: plotted data represent background-subtracted test line intensity of two replicates of independent biological duplicates and their mean.

found that the dual-epitope peptide reporter enabled five-plex detection of RNA targets following an ambient-temperature paired ligation and cell-free expression assay. The modularity of this approach could enable a rapid assay redesign to detect alternative RNA targets by only redesigning the target-binding portion of the probes.

## Sensitive detection of SARS-CoV-2 enabled by rolling circle amplification

Having established that production of a dual-epitope peptide can provide a visual multiplexed readout of INSPECTR, we next sought to improve the sensitivity of the assay. We did not think that further optimization of the transcription and translation steps could yield a substantial improvement in sensitivity. It would be necessary to increase the number of expression cassettes before the initiation of cell-free expression. Several isothermal methods of nucleic acid amplification have been demonstrated at or below 37 °C (refs. 47,48), including some that are non-instrumented and use the operator's body heat for thermal regulation[8] or run at room temperature[49]. We were interested in adopting a method that could be run at ambient temperature (22 °C) without any heat source.

We elected to optimize an amplification method on the basis of rolling circle amplification (RCA) as the primary enzymes (SplintR ligase and Φ29 DNA polymerase) are active at temperatures near ambient and amplification at or below 30 °C has been demonstrated[50].

This required the ligation probes to be re-designed and connected such that the intact expression cassette following ligation is circular rather than linear, as illustrated in Fig. 6a. Once the ligation is complete, primers and Φ29 polymerase are introduced to amplify the expression cassette. We measured the sensitivity of the process using either a single primer (linear RCA), pairs of target-specific primers (hyperbranched RCA) or random primers. We found that random 6-mer primers modified with phosphorothioate bases to prevent exonuclease digestion by Φ29 provided the best signal. We also found that the inclusion of an exonuclease step following ligation and before the RCA was essential to reduce background signal. Further, we determined that the amplification reaction was highly sensitive to glycerol concentration, so we employed high-concentration enzymes or glycerol-free formulations wherever possible.

The product of the RCA reaction starts as a long linear molecule consisting of concatemers of the circular sequence, and this enabled us to implement an additional proofreading feature. To prevent the expression of circles generated by the spurious ligation of the epitope-encoding probe to itself, the T7 promoter was moved onto the GFO. This decreased the likelihood of non-templated ligation leading to an active expression cassette because two ligation events across two different molecules are required to generate an amplifiable circle that includes both the T7 promoter and the epitope-encoding probe (Fig. 6a).

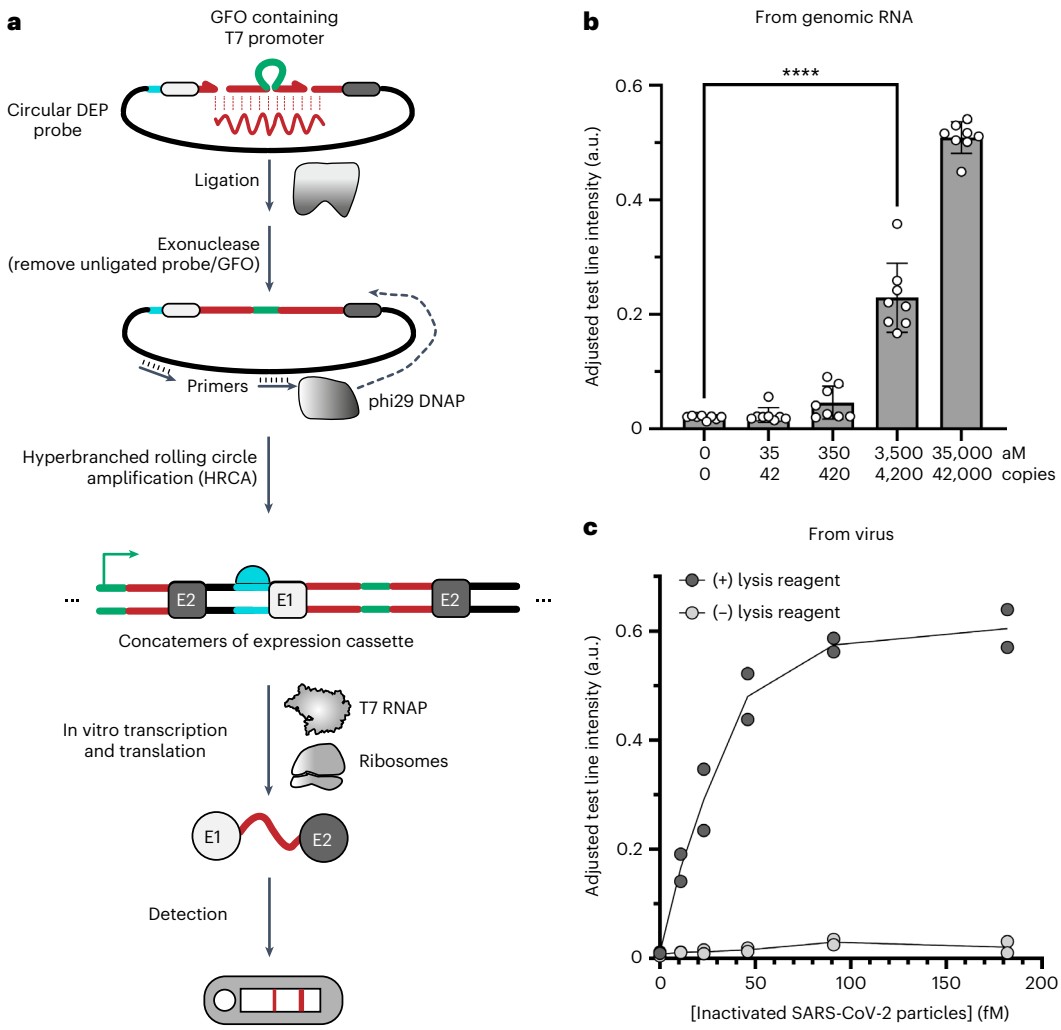

**Fig. 6 | RCA enables sensitive detection of target. a**, Modification of probes to create circular expression cassettes. By joining the two probes containing epitope sequences, the result of a target-mediated ligation event is a circular expression cassette. The addition of Φ29 and random hexamers allows the amplification of this circle. Instead of each target molecule leading to a single expression cassette as in Fig. 4a, each target is converted into thousands of expression cassettes that are further amplified by transcription and translation in the cell-free extract. **b**, Sensitivity of assay on a structured genomic RNA (gRNA) target. The RCA-assisted assay was run on a dilution series of gRNA extracted from SARS-CoV-2. As low as 3.5 fM (~4,000 cps) could be detected on a lateral-flow strip. Plotted data points represent background-subtracted test line intensity of $N = 8$ biological replicates. Error bars indicate standard deviation of the mean. ****$P < 0.0001$ for an unpaired two-tailed $t$-test with Welch's correction for unequal variance ($N = 8$). **c**, Detection of intact viral particles. Following the development of a rapid lysis method, intact SARS-CoV-2 could be detected with an LoD similar to that of extracted gRNA. Lysis is complete after 30 s at room temperature. Plotted data points represent background-subtracted test line intensities of independent biological replicates.

To demonstrate the sensitivity of the RCA-enhanced assay, dilutions of SARS-CoV-2 genomic RNA were prepared and served as templates for ligation of the circularizable probes. Following a 2 h RCA step and a 5 min exonuclease step, the amplified product was used as a substrate for cell-free expression, and the resulting peptides were analysed by lateral flow. As shown in Fig. 6b, this amplification step improved the LoD of the assay to 3.5 fM (4,200 copies) of target RNA, using a fully ambient temperature workflow.

As final validation of INSPECTR as a detection scheme for viral nucleic acids, we demonstrated compatibility with an end-to-end workflow that included lysis of intact viral particles. An extensive screen of chemicals and detergents yielded a formulation consisting of HCl and the detergent LAPAO that is capable of fully lysing SARS-CoV-2 virus at room temperature in less than 30 s, without impacting any of the downstream processes. Gamma-irradiated SARS-CoV-2 was serially diluted into water and either treated with lysis reagent or placed directly into the ligation, amplification and expression reactions. As shown in Fig. 6c, the stock of intact viral particles does not contain

accessible RNA and does not lead to any signal on the lateral-flow strip. With the inclusion of the lysis step, the assay can detect 11 fM of virus in the ligation reaction.

## Discussion

INSPECTR is an approach for developing programmable molecular diagnostics that are low-cost and compatible with instrument-free workflows. The core technology is the target-mediated ligation of DNA probes that, once joined, form an expression cassette that is transcribed and translated in a cell-free expression system to produce a desired reporter. The probes can be designed to produce any desired peptide or protein output. The selection of readout is therefore chosen according to the desired characteristics of the assay, independent of the nucleic acid target of interest.

In the simplest embodiment of INSPECTR, the probes encode a fragment of a catalytic enzyme (LacZ) that leads to a qualitative visible colour change, easily interpretable by an end user (Extended Data Fig. 2). Although the resulting colorimetric output is visible without

instrumentation, its linear operating range is narrow, and the limit of detection is high without amplification. We then developed a quantitative INSPECTR assay for use in settings that allow for a luminescence detector. Our nanoluciferase assay has a linear range ($R^2 = 1$) covering four orders of magnitude of target concentration in a diagnostically relevant range, from 1 fM to 10 pM (Fig. 3). We found that the sensitivity of the assay was sufficient to detect SARS-CoV-2 virus in human saliva (Supplementary Fig. 3), although the linearity of the assay response was impacted by the raw matrix.

INSPECTR enables a unique multiplexing capability, achieved by expressing dual-epitope peptides rather than enzymes. Following the development of a library of dual-epitope peptides and detection antibodies, we showed that a highly multiplexed lateral-flow strip can be used to detect sequences from five respiratory viruses simultaneously in a single assay. Development of additional dual-epitope peptide and detection antibodies may enable higher-order multiplexing beyond the 5-plex assay demonstrated in this work. Additional benefits of inert peptides are that they are small, easily synthesized and lack any functional domains that could be disrupted by the insertion of amino acid sequence corresponding to the target, making them easily customizable. However, as these peptides are inert, they lack the capacity for signal amplification and do not match the sensitivity of catalytic enzymes. To address this, we showed that INSPECTR can be coupled to an upstream ambient temperature amplification step to improve sensitivity, enabling the detection of 3.5 fM of SARS-CoV-2 genomic RNA (4,000 RNA copies) with a lateral-flow readout. While reports of the analytical sensitivity of molecular and antigen-based diagnostic assays vary widely on the basis of the test manufacturer and on the particular study[2,51], it can be stated that the sensitivity of INSPECTR falls between that of rapid antigen tests (cheap, but not sensitive) and PCR (very sensitive but expensive and highly instrumented) tests. Importantly, in all cases the entire process can be run at ambient temperature.

We have shown that the INSPECTR technology enables a suite of capabilities including visual readout, quantification, high sensitivity and multiplexing. The modular approach and the uncoupling of the target nucleic sequence from the amino acid output sequence make INSPECTR a highly flexible strategy for decentralized diagnostics. The use of cell extracts in INSPECTR provides several advantages. These ultra-low-cost biochemical reagents are readily programmable, can be stabilized by freeze-drying for long-term storage (greater than 1 yr) without refrigeration[52] and have been shown to function in complex matrices including saliva, urine and blood[25,53,54]. They retain significant activity at room temperature and when dried and embedded into low-cost materials such as paper[20]. However, none of the demonstrated outputs can achieve all of these capabilities simultaneously, and the number of user steps and time to result remain obstacles to their adoption at the point of need. Attempts to condense the number of user steps (that is, one-pot ligation and cell-free expression) have been hampered by incompatibility between the individual biochemical reactions (Extended Data Fig. 3). We have also observed that decreasing the duration of any individual reaction results in lower assay sensitivity (Supplementary Fig. 4). Although the current INSPECTR technology operates without the need for instrumentation and has sensitivity superior to most rapid antigen tests, further effort is required to realize a mature diagnostic solution that achieves PCR-level sensitivity and antigen-level ease of use, to enable its use by untrained users at the point of need. A summary table of the advantage and disadvantages of each explored reporter protein is shown in Supplementary Table 1.

Although we have focused on the detection of viral RNA targets to prove these concepts, the method works equally well on denatured DNA targets (Extended Data Fig. 2). The utility of INSPECTR need not be limited to infectious disease diagnostics, either. It could also be used to produce a therapeutic response—for example, the expression of antimicrobial peptides triggered by the detection of bacterial infection. Given the ability of commonly used ligases to differentiate between nucleotides at the ligation junction[42,55], INSPECTR would be useful for single-nucleotide polymorphism detection in the context of cancer diagnostics. We believe that this new implementation of synthetic biology and cell-free systems may find broad applications wherever low-cost and decentralized nucleic acid sensing is of value.

## Methods

### Cell-free extract preparation

Cell-free extract was prepared in-house as previously described[38]. Glycerol stocks (1 ml) of the indicated strain were used to inoculate a 1 l culture of either defined media, for BL21 Star (DE3) extract prep (1.5 g l$^{-1}$ ammonium sulfate, 4.6 g l$^{-1}$ potassium phosphate dibasic, 1 g l$^{-1}$ potassium chloride, 3 g l$^{-1}$ trisodium citrate dihydrate, 1 g l$^{-1}$ L-asparagine, 1.4 g l$^{-1}$ L-glycine, 0.27 g l$^{-1}$ L-histidine hydrochloride monohydrate, 0.7 g l$^{-1}$ L-isoleucine, 0.7 g l$^{-1}$ L-leucine, 0.6 g l$^{-1}$ L-lysine hydrochloride monohydrate, 0.28 g l$^{-1}$ L-methionine, 0.28 g l$^{-1}$ L-phenylalanine, 0.92 g l$^{-1}$ L-proline, 0.73 g l$^{-1}$ L-threonine, 0.28 g l$^{-1}$ L-tryptophan, 0.34 g l$^{-1}$ L-tyrosine, 0.45 g l$^{-1}$ L-valine, 0.65 g l$^{-1}$ betaine hydrochloride, 0.02 g l$^{-1}$ ferric chloride hexahydrate, 28.6 mg l$^{-1}$ choline chloride, 25.1 mg l$^{-1}$ niacin, 25.6 mg l$^{-1}$ $p$-aminobenzoic acid potassium salt, 9.4 mg l$^{-1}$ pantothenic acid hemicalcium salt, 1.5 g l$^{-1}$ pyridoxine hydrochloride, 3.9 mg l$^{-1}$ riboflavin, 17.7 mg l$^{-1}$ thiamine hydrochloride, 0.1 mg l$^{-1}$ biotin, 0.1 mg l$^{-1}$ cyanocobalamin, 0.07 mg l$^{-1}$ folic acid dihydrate, 3.5 mg l$^{-1}$ sodium molybdate, 3.9 mg l$^{-1}$ zinc sulfate heptahydrate, 1.2 mg l$^{-1}$ boric acid, 3.4 mg l$^{-1}$ cobalt (II) chloride hexahydrate, 3.4 mg l$^{-1}$ copper (II) sulfate pentahydrate, 1.9 mg l$^{-1}$ manganese (II) sulfate, 0.2 g l$^{-1}$ magnesium sulfate, 5 g l$^{-1}$ glucose, 5 g l$^{-1}$ yeast extract and hydrochloric acid to pH 7.2) or 2X YTP + G (for DH10β, 16 g l$^{-1}$ tryptone, 10 g l$^{-1}$ yeast extract, 5 g l$^{-1}$ sodium chloride, 7 g l$^{-1}$ potassium phosphate dibasic, 3 g l$^{-1}$ potassium phosphate monobasic, 5 g l$^{-1}$ glucose).

Cultures were inoculated in a 2.5 l Tunair flask to a starting optical density (OD$_{600}$) of 0.2. Cultures were incubated on a shaker at 37 °C and 225 r.p.m. until they reached an optical density between 4 and 5. The flask was chilled on ice water for 30 min, then centrifuged at 6,000 × $g$ for 30 min at 10 °C to pellet the cells. The pellets were fully resuspended by vortexing in 30 ml S30 buffer (10 mM Tris acetate, 14 mM magnesium acetate, 60 mM potassium acetate, pH 8.2). The suspension was re-centrifuged at 6,000 × $g$ for 30 min at 4 °C and the supernatant removed. The 30 ml wash was repeated. After the third centrifugation, the pellets were flash frozen in liquid nitrogen and stored overnight or until use. To prepare extracts, the pellets were thawed on ice in 1.25 ml S30 buffer per gram cell pellet and fully resuspended. The cells were lysed by homogenization on an Avestin EmulsiFlex-C3 homogenizer and pre-equilibrated with S30 buffer using three passes at 10–15,000 psi. The lysate was centrifuged at 18,000 × $g$ for 30 min at 4 °C for clarification. The supernatant was removed and incubated at 37 °C, shaking at 140–200 r.p.m. for a ribosomal run-off reaction for 30 min. The clarified extract was the supernatant of one additional centrifugation at 18,000 × $g$ for 30 min at 4 °C, was flash frozen on liquid nitrogen and stored at −80 °C for long-term storage.

### Cell-free gene expression

The base CFE reactions using commercial cell-free expression kits were run with the following compositions:

NEBExpress: 12.5% S30 synthesis extract and 25% Protein Synthesis buffer (E5360, New England Biolabs (NEB)), 9 U μl$^{-1}$ T7 RNAP (M1019B, NEB), 0.8 U μl$^{-1}$ murine RNase inhibitor (M0314S, NEB).

PURExpress: 28% solution A, 21% solution B (E6800, NEB), 0.8 U μl$^{-1}$ murine RNase inhibitor (M0314S, NEB).

The base CFE reactions using in-house lysate were run at the following composition: 12.5% extract (v/v), 0.6 mM adenosine triphosphate (AAJ6033622, Thermo Fisher), 0.6 mM GTP (AAJ1680003, Thermo Fisher), 0.15 mM cytidine triphosphate (AAJ62238-03, VWR), 0.15 mM uridine triphosphate (ICN19123091, Thermo Fisher); 34 μg ml$^{-1}$ folinic acid (47612, Sigma), 170 μg ml$^{-1}$ $E.$ $coli$ tRNA (10109550001, Sigma),

2 mM each amino acid (VWR), 33.33 mM phosphoenolpyruvate (AAB2035822, Thermo Fisher), 0.33 mM NAD (AAJ6233706, Thermo Fisher), 0.27 mM CoA (102588-436, VWR), 2.7 mM potassium oxalate (223425, Sigma), 1 mM putrescine (D13208, Sigma), 1.5 mM spermidine (AAAA19096-22, VWR), 10 mM ammonium glutamate, 10 mM magnesium glutamate (49605, Sigma), 105 mM potassium glutamate (AAA17232-30, VWR), 27 U μl$^{-1}$ T7 RNAP (M1019B, NEB), 30 μg ml$^{-1}$ GamS nuclease inhibitor (P0774S, NEB), 0.8 U μl$^{-1}$ murine RNase inhibitor (M0314S, NEB) and the indicated concentration of DNA.

For the measurement of LacZ reporter, additionally 0.75 mM chlorophenol red-$\beta$-D-galactopyranoside (59767, Sigma) and 55 nM LacZ omega fragment (P2010005, Molecular Depot) were added to 30 μl reactions, which were continuously monitored for absorbance at 576 nm every minute on a BioTek Gen5 (v3.11 software) plate reader on a sealed (AB0558, Thermo Fisher) black clear-bottom plate (3764, Corning). The reported time to absorbance = 1 was computed by linear interpolation. DH10β was used as the source strain to prepare extract as it is deficient in full-length genomic LacZ, which would otherwise lead to very high background.

For the measurement of nLuc reporter in lysate, 1% v/v Nano-Glo luciferase substrate (N1110, Promega) was supplemented to the reactions, along with 10% polyethylene glycol (PEG)-8000 (1546605, Sigma) and 20% PURExpress solution B (to enhance translation), which further enhanced the reaction sensitivity. Reported data are the highest reported luminescence over the course of a 2 h experiment run at room temperature and 10 μl scale, continuously monitored every minute on a sealed 384-square-well white plate (4513, Corning). Protein production was estimated from a linear calibration curve using purified nLuc. BL21 Star (DE3) was the source strain for expression of nLuc in Fig. 1. For the HiBiT experiments in Supplementary Figs. 3 and 6, 0.5 vol% LgBiT purified subunit was additionally used (Promega N3030).

### Screening for functional dsDNA nLuc variants
gBlock gene fragments encoding the nLuc variants were ordered from Integrated DNA Technologies (IDT). Linear expression templates for the variants were generated using PCR primers annealing to the T7 promoter and T7 terminator of the expression cassettes, and the PCR products were purified using PureLink PCR purification kit (K310001, Invitrogen) and quantified by Take3 assay. Variants were screened by incubating 2 fM of the diluted linear expression template in a 10 μl reaction volume using in-house BL21 Star (DE3) extract.

### Quantitative detection of SARS target
ssDNA probes A (NG15) and B (NG16) were ordered from GenScript. Ligation reactions were performed in a 5 μl reaction volume containing 1x T4 RNA ligase buffer (B0216L, NEB), 7% v/v PEG 3350, 1 U μl$^{-1}$ murine RNase inhibitor (M0314L, NEB), 65 ng μl$^{-1}$ extreme thermostable single-stranded DNA binding protein (ET SSB; M2401S, NEB), 840 nM SplintR ligase (M0375S, NEB), 9 mM dithiothreitol, 500 nM reverse primer (to initiate double-stranding from the 3′ terminus), 3.8 nM probe A and 380 pM probe B. The linear probe concentrations were chosen to maximize signal and reduce leak; increasing probe B but not probe A to 3.8 pM led to higher leak, and we maximized probe concentration where possible. Reactions were initiated by the addition of SARS-CoV-2 genomic RNA or synthetic RNA at the indicated concentrations and then incubated at ambient temperature for 30 min. Following ligation, a 10 μl CFE reaction master mix containing 40% v/v of PureExpress solution A, 30% PureExpress solution B (E6800L, NEB), 0.38 U μl$^{-1}$ IsoPol (71500-201, ArcticZymes), 1.5x Nano-Glo substrate (PRN1110, Promega), 0.15 mM deoxynucleotide triphosphates (dNTPs) and 7.5 mM EDTA were directly added to the 5 μl ligation reaction. Luminescence was measured on a BioTek Gen5 (v3.11 software) plate reader at ambient temperature for 20 min and the highest measured value is reported. The nonlinear ordinary least-squares fit was computed on the log-transformed x and y values in the range 10–10$^5$ fM target RNA.

The output model (log LUMINESCENCE) = m1 log (RNA/fM) + b1) gave 95% CI parameters m1 = (1.102,1.125), b1 = (0.004555,0.1202) and $R^2$ = 1.000 from 15 data points. We also computed a linear-linear model across the full domain, where LUMINESCENCE = m2 (RNA/fM) + b2, with 95% CIs m2 = (0.9381,0.9493), b2 = (1.938,6.541) and $R^2$ = 0.9999.

For the detection of clinical samples in Supplementary Fig. 5, the saliva samples were pre-treated with 20 mM NaOH solution for 30 s and then neutralized in the ligation reaction with a 4 mM HCl solution. Following this, the ligation reactions were allowed to proceed as described above.

### SARS genomic RNA or synthetic RNA
SARS synthetic RNA fragments were ordered from IDT and full sequence synthetic RNA reconstituted by pooling the RNA fragments. SARS genomic RNA was purified from viral particles cultured in Vero cells (0810587CFHI, Zeptometrix) using the PureLink Viral DNA/RNA kit (12280050, Invitrogen) following manufacturer protocol.

### Lateral-flow strip preparation
To prepare the detection antibody conjugate, anti-*Strep*-tag II antibody (clone 5A9F9, A01732, Genscript) was buffer exchanged into 10 mM potassium phosphate, pH 7.4 (DSKR, NanoComposix). Five ml of 150 nm carboxyl gold nanoshells (20 OD) were activated with 400 μg of 1-ethyl-3-(-3-dimethylaminopropyl) carbodiimide hydrochloride followed by 800 μg of sulfo-*N*-hydroxysuccinimide and then resuspended into reaction buffer (5 mM potassium phosphate buffer with 0.5% PEG (20 kDa, pH 7.4)). Buffer-exchanged antibody (75 μg) was diluted 4X into reaction buffer and incubated with activated nanoshells for 1 h. After antibody coupling, the conjugate was blocked with blocking buffer (0.5X PBS, 5% BSA, 0.5% casein, 1% Tween 20, 0.05% sodium azide, pH 8) for 1 h. Blocked conjugate was washed with reaction buffer and then resuspended to 20 OD with conjugate diluent buffer (0.5X PBS, 0.5% BSA, 0.5% casein, 1% Tween 20, 0.05% sodium azide, pH 8).

The universal lateral-flow strips (Figs. 2b, 4a and 6) were prepared by laminating an absorbent pad (CF5, Cytiva) onto a 22-mm-wide nitrocellulose membrane (FF120HP, Cytiva) and backing card (KN-PS1045. DEV, Kenosha Tapes). Using a reagent dispenser (XYZ3060, Biodot), antibodies were striped at a rate of 1 μl cm$^{-1}$: the anti-mouse control line (0.5 mg ml$^{-1}$ in PBS, A28174, Thermo Fisher) was striped 17 mm from the bottom of the nitrocellulose, and an anti-FLAG test line (1 mg ml$^{-1}$ in PBS, clone M2.1, cAb6404-1.1, Absolute Antibody) was striped 12 mm from the bottom of the nitrocellulose. The 7-plex test strips (Fig. 4b) were similarly prepared by laminating the absorbent pad onto a 45-mm-wide nitrocellulose membrane and backing card. Anti-HA (clone 5E11D8, A01244-100, Genscript), anti-Ty1 (clone BB2, MA5-23513, Thermo Fisher) and anti-OLLAS (clone L2, NBP1-0673, NovusBio) antibodies were resuspended to 0.5 mg ml$^{-1}$ in PBS; anti-V5 (clone SV5-Pk1, R960-25, Invitrogen), anti-S-tag (clone GT727, SAB2702227, Sigma Aldrich), anti-VSV-G (clone P5D4, ab50549, Abcam) and anti-Myc (clone 9E10, M4439, Sigma Aldrich) antibodies were buffer exchanged into a striping buffer (0.1 M sodium phosphate buffer, pH 8 with 2% trehalose and 5% sucrose). The antibodies were striped 5 mm apart at 1 μl cm$^{-1}$ in the following striping order: anti-mouse (C), anti-Ty1 (T7), anti-HA (T6), anti-VSV-G (T5), anti-S-tag (T4), anti-Myc (T3), anti-V5 (T2) and anti-OLLAS (T1). For the 5-plex test strip (Fig. 5), antibodies were striped in the following order: anti-mouse (C), anti-FLAG (T5), anti-Ty1 (T4), anti-VSV-G (T3), anti-S-tag (T2) and anti-V5 (T1). Striped membranes were dried at 37 °C for 1 h and cut into 4-mm-wide strips with a guillotine cutter (CM5000, Biodot).

### Dual-epitope peptide expression and detection
To evaluate the universal detection of a 3x-FLAG (N-terminus) and Twin-Strep-tag (C-terminus) dual-epitope reporter (Figs. 2b and 4a), 10 pM of expression cassette (various dsDNA gBlocks, IDT) was spiked into 10 μl PURExpress (Figs. 2b and 4a) and in-house extract derived

from BL21 Star (DE3) (Fig. 2b) expression reactions. Expression reactions were incubated at 22 °C for 2 h. Post expression, all reaction products were then diluted with 25 µl phosphate-buffered saline with Tween-20 (PBST) and 5 µl of detection conjugate to a final volume of 40 µl into a non-binding, flat-bottomed 96-well plate. A universal lateral-flow strip was then deposited into the reaction; a visible test line indicated a sandwiched capture of the dual-epitope peptide with the anti-FLAG test line and anti-*Strep*-tag II detection antibody conjugate.

To evaluate multiplexed detection of 7 dual-epitope peptides (Fig. 4b), a 30 µl expression reaction composed of 0.73X S30 Synthesis extract (P0864B, NEB), 0.73X Protein Synthesis buffer (B0864B, NEB), 13.2 U µl⁻¹ T7 polymerase (M1019B, NEB), 1.17 U µl⁻¹ RNAse inhibitor (M1018B, NEB) and 143 pM expression cassette (custom gBlocks, IDT) was incubated at 22 °C for 2 h. The expressed product (27 µl) was mixed with 5 µl of detection conjugate and PBST to a final volume of 40 µl. A 7-plex lateral-flow strip was then deposited into the reaction; visible test lines indicated a sandwiched capture of the dual-epitope peptide with peptide-specific capture test lines and anti-*Strep*-tag II detection antibody conjugate.

To evaluate multiplexed detection of 5 RNA targets (Fig. 5), target-specific probes were ligated together to initiate the formation of expression cassettes. Ligation reactions (12 µl) contained 4% w/v PEG 3350 (1008055, Rigaku Reagents), 1 U µl⁻¹ RNAse inhibitor (M1018B, NEB), 0.0325 µg µl⁻¹ ET SSB (M2401B-HC1, NEB), 1X SplintR ligase buffer (B0375S, NEB), 0.84 µM SplintR ligase (M0375B-HC1, NEB), 0.5 µM reverse primer (Supplementary Table 1, IDT) and 10 nM of each target RNA. The ligation reactions were incubated for 30 min at 22 °C. The ligation product (10 µl) was then added to a final expression reaction volume of 30 µl containing 0.73X S30 Synthesis extract (P0864B, NEB), 0.73X Protein Synthesis buffer (B0864B, NEB), 13.2 U µl⁻¹ T7 polymerase (M1019B, NEB), 1.17 U µl⁻¹ RNAse inhibitor (M1018B, NEB), 100 µM dNTPs (N0447L, NEB) and 33.3 U ml⁻¹ Bsu DNA polymerase (M0330L, NEB), and incubated at 22 °C for 2 h. The expressed product (27 µl) was mixed with 5 µl of detection conjugate and PBST to a final volume of 40 µl. A 5-plex lateral-flow strip was then deposited into the reaction; visible test lines were expected to indicate a sandwiched capture of the dual-epitope peptide with peptide-specific capture test lines and anti-*Strep*-tag II detection antibody conjugate.

### Sensitive detection of SARS-CoV-2 enabled by RCA

For each genomic RNA concentration, a 7 µl ligation reaction was prepared using 840 nM SplintR ligase (M0375B-HC1, NEB), 1X SplintR reaction buffer, 32.4 ng µl⁻¹ ET SSB (M2401B-HC1, NEB), 1 U µl⁻¹ murine RNase inhibitor (M0314L, NEB), 4% PEG 3350 (1008055, Rigaku Reagents), 1.5 nM circularizable and 5′-phosphorylated ssDNA probe, 15 nM 5′-phosphorylated GFO and 2 µl of the indicated concentration of SARS gRNA. The reaction was incubated at 22 °C on a thermal cycler for 30 min. Exonuclease I (M0293B-HC1, NEB) was then added to a final concentration of 2.5 U µl⁻¹, and the reaction was further incubated at 22 °C for 5 min.

RCA reactions were prepared in a total volume of 12 µl consisting of 6 µl ligation product, 1 mM dNTPs (N0447L, NEB), 1X Φ29 buffer, 1 U µl⁻¹ Φ29 DNA polymerase (M0269B-HC1, NEB) and 40 µM random hexamer RCA primers with 2X phosphorothioate linkages at the 3′ terminus. The reaction was incubated at 22 °C for 2 h.

Expression of the amplified RCA product was done using NEB-Express Cell-free *E. coli* Protein Synthesis system (E5360L, NEB) in a total volume of 30 µl, consisting of 10 µl of the RCA product, 0.73X S30 Synthesis extract (P0864B, NEB), 0.73X Protein Synthesis buffer (B0864B, NEB), 13.2 U µl⁻¹ T7 polymerase (M1019B, NEB), 1.17 U µl⁻¹ RNAse inhibitor (M1018B, NEB), 1.2 U µl⁻¹ murine RNase inhibitor and 0.5 mg ml⁻¹ salmon sperm DNA (15632011, Invitrogen), and incubated at 22 °C for 2 h.

To visualize the dual-epitope peptides, 27 µl of the expressed product was mixed with 5 µl of detection conjugate and PBST to a final

volume of 40 µl and applied to a lateral-flow strip. A two-tailed *t*-test on unpaired means using Welch's correction for unequal variances was conducted between the 0 and 3,500 aM conditions (*N* = 8 at each condition) and obtained *t* = 9.770 and adjusted d.f. = 7.045 (*P* < 0.0001).

To demonstrate the detection of intact viral particles, the virus used was gamma-irradiated SARS-CoV-2 (NR-52287, BEI Resources). For the no-lysis condition, each concentration of virus was added directly to the ligation reaction as described above. For the lysed samples, each concentration was first pre-treated by incubating in a lysis reagent consisting of 20 mM HCl, 0.1% LAPAO (L360S, Anatrace) and 1 U µl⁻¹ RNAsin (N2511, Promega) for 30 s at room temperature, and then neutralized by the addition of 100 mM Tris-Cl (pH 9.0) before being added to the ligation mix. The workflow then proceeded exactly as described above for the gRNA detection.

A step-by-step protocol describing the assay can be found at Protocol Exchange with the title 'INSPECTR assay protocol for sensitive SARS-CoV-2 enabled by rolling circle amplification'.

### Lateral-flow strip readout

All test strips were developed for 20 min and then scanned (V850 Pro scanner, Epson); the mean grey values of test lines and background (25 pixels below the test line) were measured by ImageJ 1.53t. The resultant 'adjusted test line intensity' was then calculated as

$$I_{\text{adjusted}}\ \text{test line} = \frac{I_{\text{background}} - I_{\text{raw}}}{I_{\text{background}}}$$

where 0 represents the lowest possible pixel intensity (that is, black).

### Reporting summary

Further information on research design is available in the Nature Portfolio Reporting Summary linked to this article.

## Data availability

All data needed to evaluate the conclusions can be found in the paper and its Supplementary Information. Source data are provided with this paper.

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

## Acknowledgements

We thank I. Azmi, K. Cholkar and D. Paszkowski for producing the in-house cell-free extract; H. Boisvert for assistance with optimizing protein expression and for input during manuscript writing; L. Carlson for assistance with optimizing protein expression; M. Wilson for screening-assay components and for input on assay design and manuscript writing; J. Lo, S. Singh and M. Desmond for assistance on the screening-assay components; J. R. Swartz at Stanford University for guidance on the production of cell-free extract and on protein expression; M. B. Frieman at the University of Maryland for providing SARS genomic RNA for early INSPECTR experiments. Sherlock Biosciences discloses support for the research described in this study from the Good Venture Foundation and the Bill and Melinda Gates Foundation (grant number 022787).

## Author contributions

E.A.P. and A.D.S. conceived and designed the experiments, performed the experiments, analysed the data and wrote the manuscript. A.J. conceived and designed the experiments, analysed the data and wrote the manuscript. C.B. and P.C. conceived and designed the experiments, performed the experiments, analysed the data and edited the manuscript. M.L., C.C., K.S., N.G., J.H., J.E.H., E.R. and J.A. planned and performed experiments and analysed the data. A.L., V.C. and P.R. designed experiments and analysed the data. J.J.C. and W.J.B. conceived and designed the experiments.

## Competing interests

All authors except J.J.C. are currently employed or were previously employed by Sherlock Biosciences. J.J.C. is a co-founder of Sherlock Biosciences.

## Additional information

**Extended data** is available for this paper at https://doi.org/10.1038/s41551-023-01028-y.

**Correspondence and requests for materials** should be addressed to Aric Joneja.

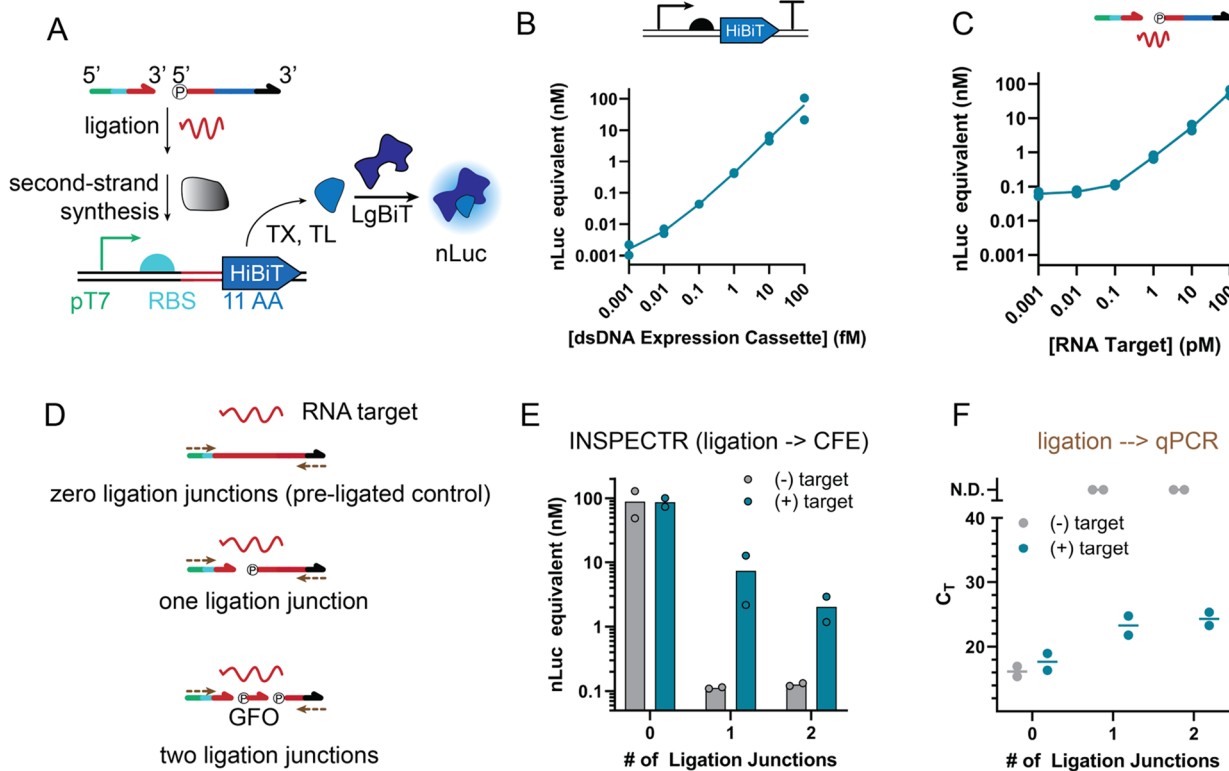

**Extended Data Fig. 1 | Gap-filling oligos reduce the efficiency of splint-ligation by about 50%. a**, Schematic of the HiBiT probe used in this study. We chose this expression system as a model for ligation efficiency because the probe has a similar structure and length as the dual-epitope peptide probes, but it provides a quantitative readout, similar to nanoluciferase[56]. **b**, Expression cassette titration for dsDNA encoding the ligated HiBiT probe. Plotted data (N = 2) represent the individual points and average of 2-hour endpoint luminescence after expression in in-house produced BL21 Star (DE3) extract. **c**, Target titration for RNA on the HiBiT probe. A SARS-CoV-2 targeting probe was designed with the ligation junction in the 5′ UTR, since the HiBiT peptide itself is

only 11 AAs. Ligation was performed for 30 minutes, followed by two hours of cell-free expression. **d**, Schematic of HiBiT probes with 0 (pre-ligated), 1, or 2 ligation junctions (including the gap-filling oligo). The junctions were designed to occur within the coding sequence of the peptide, so that any un-ligated products would not show detectable expression leak. **e**, The addition of a second splint-junction for a single target RNA decreases the signal from a quantitative HiBiT probe around 2-fold and **f**, increases the concentration of ligated target by about one $C_T$, when measured by quantitative PCR. Probes were ligated for 30 minutes in the presence or absence of 1 nM target RNA.

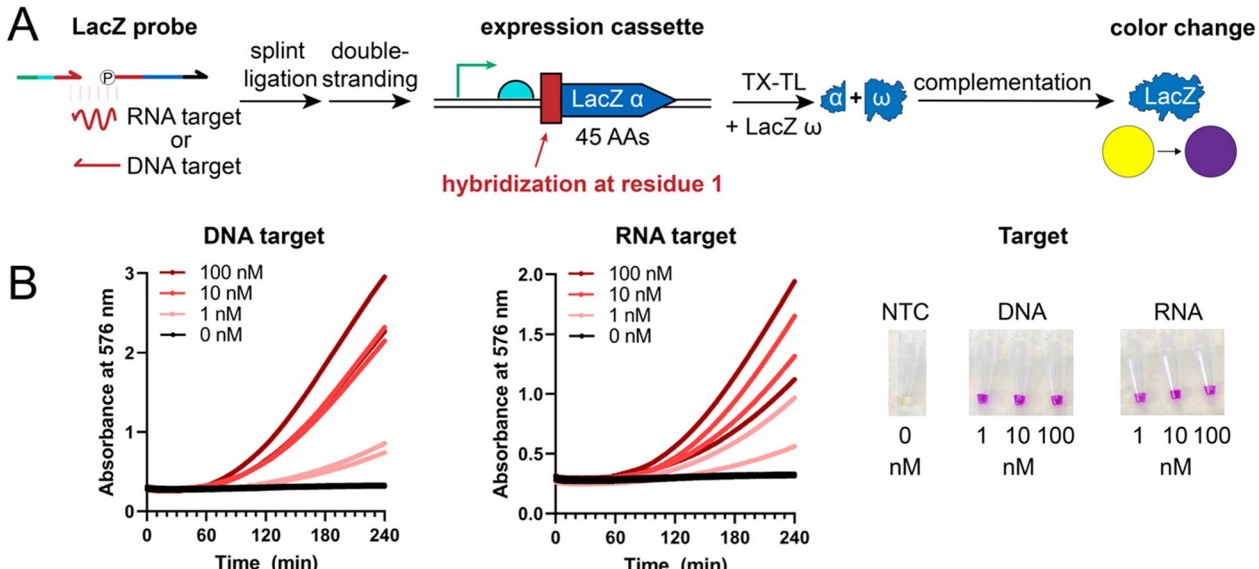

**Extended Data Fig. 2 | Design and implementation of a colorimetric INSPECTR probe for RNA or DNA detection. a**, Schematic of the colorimetric INSPECTR reaction. Probes 'A' and 'B' were split at a hybridization region targeting SARS-CoV-2 sense sequence. To avoid synthesizing the full-length LacZ gene on a single-stranded probe, the probe was designed instead to only express the LacZ alpha fragment, which complements with the purified omega fragment to produce the intact enzyme, as previously described. Because the LacZ alpha peptide is small, we did not split it and instead placed the hybridization region immediately after the start codon, in the same reading frame as the coding sequence. (We found that this design did not show detectable transcriptional leak.) **b**, Detection of SARS-CoV-2 nucleic acid sequences using colorimetric INSPECTR probes. 10 nM of LacZ probe was splint-ligated on either SARS-CoV-2 RNA sequence or the corresponding DNA at the indicated concentration; then, 3 µL was added to 7 µL CFE reactions using cell-free extract prepared from DH10β *E. coli* cells, along with purified LacZ omega-fragment and chlorophenol red galactopyranoside (the reaction substrate). Left: kinetics of expression and complementation from N = 2 independent ligation and expression reactions using the indicated concentration of DNA or RNA targets. Right: representative images of the reaction endpoints (18 hours) taken by iPhone.

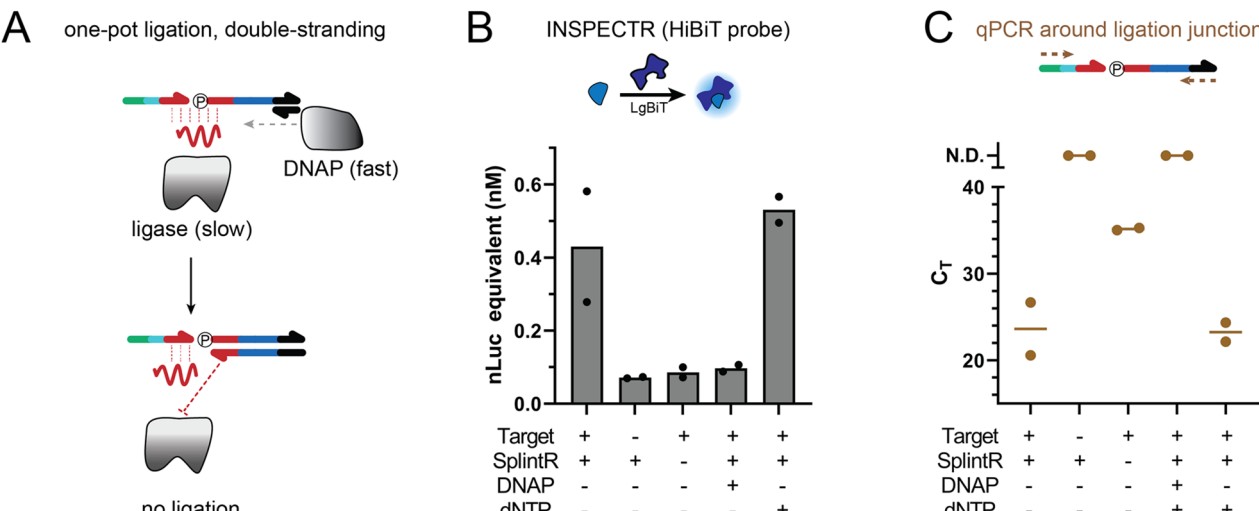

**Extended Data Fig. 3 | One-pot INSPECTR fails due to the incompatibility in reaction rates between splint-ligation and probe double-stranding.** **a**, Schematic of one-pot ligation-second strand synthesis. If second strand synthesis is faster than the ligase, then the single-stranded junction won't be accessible for SplintR ligase. Measured both by **b**, cell-free expression and **c**, qPCR using primers that span the ligation junction, as well as cell-free expression, the addition of exogenous DNAP, which is necessary to generate a transcribable expression cassette, impedes the splint-ligation of functional probe.

# Reporting Summary

## Statistics

For all statistical analyses, confirm that the following items are present in the figure legend, table legend, main text, or Methods section.

| n/a | Confirmed | |
|---|---|---|
| ☐ | ☒ | The exact sample size (*n*) for each experimental group/condition, given as a discrete number and unit of measurement |
| ☐ | ☒ | A statement on whether measurements were taken from distinct samples or whether the same sample was measured repeatedly |
| ☐ | ☒ | The statistical test(s) used AND whether they are one- or two-sided<br>*Only common tests should be described solely by name; describe more complex techniques in the Methods section.* |
| ☒ | ☐ | A description of all covariates tested |
| ☐ | ☒ | A description of any assumptions or corrections, such as tests of normality and adjustment for multiple comparisons |
| ☐ | ☒ | A full description of the statistical parameters including central tendency (e.g. means) or other basic estimates (e.g. regression coefficient) AND variation (e.g. standard deviation) or associated estimates of uncertainty (e.g. confidence intervals) |
| ☐ | ☒ | For null hypothesis testing, the test statistic (e.g. *F*, *t*, *r*) with confidence intervals, effect sizes, degrees of freedom and *P* value noted<br>*Give P values as exact values whenever suitable.* |
| ☒ | ☐ | For Bayesian analysis, information on the choice of priors and Markov chain Monte Carlo settings |
| ☒ | ☐ | For hierarchical and complex designs, identification of the appropriate level for tests and full reporting of outcomes |
| ☒ | ☐ | Estimates of effect sizes (e.g. Cohen's *d*, Pearson's *r*), indicating how they were calculated |

*Our web collection on statistics for biologists contains articles on many of the points above.*

## Software and code

Policy information about availability of computer code

| Data collection | Absorbance and luminescence values were collected via Biotek Neo2 with Gen 5 software (ver. 3.08). Lateral-flow test strips were imaged on an Epson V850 Pro scanner with EPSON Scan software (ver. 3.9.3.4). |
|---|---|
| Data analysis | ImageJ Version 1.53, Microsoft Excel 365 Version 2206, GraphPad Prism 9. |

For manuscripts utilizing custom algorithms or software that are central to the research but not yet described in published literature, software must be made available to editors and reviewers. We strongly encourage code deposition in a community repository (e.g. GitHub). See the Nature Portfolio guidelines for submitting code & software for further information.

## Data

Policy information about availability of data

All manuscripts must include a data availability statement. This statement should provide the following information, where applicable:
- Accession codes, unique identifiers, or web links for publicly available datasets
- A description of any restrictions on data availability
- For clinical datasets or third party data, please ensure that the statement adheres to our policy

All data needed to evaluate the conclusions can be found in the paper and its supplementary information. Source data are provided with this paper.

## Human research participants

Policy information about studies involving human research participants and Sex and Gender in Research.

| | |
|---|---|
| Reporting on sex and gender | The study did not involve human research participants. |
| Population characteristics | — |
| Recruitment | — |
| Ethics oversight | — |

Note that full information on the approval of the study protocol must also be provided in the manuscript.

# Field-specific reporting

Please select the one below that is the best fit for your research. If you are not sure, read the appropriate sections before making your selection.

☒ Life sciences ☐ Behavioural & social sciences ☐ Ecological, evolutionary & environmental sciences

For a reference copy of the document with all sections, see nature.com/documents/nr-reporting-summary-flat.pdf

# Life sciences study design

All studies must disclose on these points even when the disclosure is negative.

| | |
|---|---|
| Sample size | No sample-size calculation was performed; replicates or triplicates were deemed to be sufficient on the basis of previous observations of variability in the systems. |
| Data exclusions | No data were excluded. |
| Replication | The data provided in the paper are representative of the results recorded over many days and experiments, with similar (yet not identical) processes and outcomes. |
| Randomization | Only one variable (typically, target concentrations) was introduced into each experiment. |
| Blinding | The results of the analyses were objective and quantifiable, and not subject to investigator bias. |

# Reporting for specific materials, systems and methods

We require information from authors about some types of materials, experimental systems and methods used in many studies. Here, indicate whether each material, system or method listed is relevant to your study. If you are not sure if a list item applies to your research, read the appropriate section before selecting a response.

### Materials & experimental systems

| n/a | Involved in the study |
|---|---|
| ☐ | ☒ Antibodies |
| ☒ | ☐ Eukaryotic cell lines |
| ☒ | ☐ Palaeontology and archaeology |
| ☒ | ☐ Animals and other organisms |
| ☒ | ☐ Clinical data |
| ☒ | ☐ Dual use research of concern |

### Methods

| n/a | Involved in the study |
|---|---|
| ☒ | ☐ ChIP-seq |
| ☒ | ☐ Flow cytometry |
| ☒ | ☐ MRI-based neuroimaging |

## Antibodies

| | |
|---|---|
| Antibodies used | anti-Strep-tag IIantibody (clone 5A9F9, A01732, Genscript, lot H2012001)<br>anti-FLAG test line (clone M2.1, cAb6404-1.1, Absolute Antibody, lots T2112B21, T2112B18)<br>Anti-HA (clone 5E11D8, cat. A01244-100, Genscript, lot 19L002034)<br>anti-Ty1 (clone BB2, cat. MA5-23513, ThermoFisher, lots WK3436911, WK3234861, and WK3440671)<br>anti-OLLAS (clone L2, cat. NBP1-0673, NovusBio, lots F-15 and F-16)<br>anti-V5 (clone SV5-Pk1, cat. R960-25, Invitrogen, lot 2330339) |

anti-S-tag (clone GT727, cat. SAB2702227, Sigma Aldrich, lot 41295)
anti-VSV-G (clone P5D4, cat. ab50549, Abcam, lot GR34014830-5)
anti-Myc (clone 9E10, cat. M4439, Sigma Aldrich, lot 0000129827)
anti-mouse (polyclonal, cat. A28174, Thermo Fisher, lots 2324159 and 2352345)

Validation

Antibodies were validated by the manufacturers by ELISA. We also validated each antibody by ELISA and in a lateral-flow immunoassay format in which antibodies were striped onto a test membrane or conjugated to 40-nm gold nanoparticles. An antibody was considered valid if a test line appeared when subject to a synthesized peptide; no test line appeared if no peptide or off-target peptides were present.

