## [Peer Review File · Nature Biomedical Engineering]

Detection of viral RNAs at ambient temperature via reporter proteins produced through the target-splinted ligation of DNA probes

Corresponding author: Aric Joneja

Editorial note

This document includes relevant written communications between the manuscript's corresponding author and the editor and reviewers of the manuscript during peer review. It includes decision letters relaying any editorial points and peer-review reports, and the authors' replies to these (under 'Rebuttal' headings). The editorial decisions are signed by the manuscript's handling editor, yet the editorial team and ultimately the journal's Chief Editor share responsibility for all decisions.

Any relevant documents attached to the decision letters are referred to as **Appendix #**, and can be found appended to this document. Any information deemed confidential has been redacted or removed. Earlier versions of the manuscript are not published, yet the originally submitted version may be available as a preprint. Because of editorial edits and changes during peer review, the published title of the paper and the title mentioned in below correspondence may differ.

Correspondence

Tue 16 Aug 2022

Decision on Article nBME-22-1597

Dear Dr Joneja,

Thank you again for submitting to *Nature Biomedical Engineering* your manuscript, "Multiplexed viral RNA detection with target-splinted ligation probes forming cell-free expression cassettes". The manuscript has been seen by three experts, whose reports you will find at the end of this message.

You will see that the reviewers appreciate the work. However, they raise questions regarding the specifics of the assay's workflow and performance, and provide useful suggestions for improvement. We hope that with significant further work you can address the criticisms and convince the reviewers of the merits of the study. In particular, we would expect that a revised version of the manuscript provides:

- * Further optimization of the assay, for sensitivity and turnaround time, and discussion of additional optimization steps that could be implemented in future.
- * Evidence of assay performance with patient samples.
- * Discussion of the assay's current performance limitations and implementation constraints.
- * Thorough methodology, including a step-by-step protocol (placed in Methods or as Supplementary Information).

When you are ready to resubmit your manuscript, please upload the revised files, a point-by-point rebuttal to the comments from all reviewers, the reporting summary, and a cover letter that explains the main improvements included in the revision and responds to any points highlighted in this decision.Please follow the following recommendations:

- * Clearly highlight any amendments to the text and figures to help the reviewers and editors find and understand the changes (yet keep in mind that excessive marking can hinder readability).
- * If you and your co-authors disagree with a criticism, provide the arguments to the reviewer (optionally, indicate the relevant points in the cover letter).
- * If a criticism or suggestion is not addressed, please indicate so in the rebuttal to the reviewer comments and explain the reason(s).
- * Consider including responses to any criticisms raised by more than one reviewer at the beginning of the rebuttal, in a section addressed to all reviewers.
- * The rebuttal should include the reviewer comments in point-by-point format (please note that we provide all reviewers will the reports as they appear at the end of this message).
- * Provide the rebuttal to the reviewer comments and the cover letter as separate files.

We hope that you will be able to resubmit the manuscript within 15 weeks from the receipt of this message. If this is the case, you will be protected against potential scooping. Otherwise, we will be happy to consider a revised manuscript as long as the significance of the work is not compromised by work published elsewhere or accepted for publication at *Nature Biomedical Engineering*.

We hope that you will find the referee reports helpful when revising the work. Please do not hesitate to contact me should you have any questions.

Best wishes,

Pep

Pep Pàmies
Chief Editor, Nature Biomedical Engineering

Reviewer #1 (Report for the authors (Required)):

The manuscript by Dr Joneja and colleagues presents an instrument-free INSPECTR method for multiplexed viral RNA detection based on target-splinted ligation and cell-free expression reactions. This method is very simple and practical, relies on the target-splinted ligation of DNA probes to make cell-free expression cassettes programmed for the synthesis of reporter proteins. In this work, small 45-amino acid alpha-fragment of LacZ, nanoluciferase (nLuc), and dual epitope peptides were investigated as reporter proteins for detection of five respiratory viral targets. Following a two-hour pre-amplification of RCA, the INSPECTR method enabled the detection of 3.5 fM of SARS-CoV-2 genomic RNA (4000 RNA copies) with a lateral flow readout. The data of the manuscript are comprehensive and convincing. The work is interesting and the presentation is adequate.

However, I think that its conceptual and methodological novelty is insufficient. The methods for detect of nucleic acids based on target-induced proximity ligation followed by amplification reaction or cell-free expression reactions have been well-established in past decades. The INSPECTR method did not provide a sufficiently striking advance and potential importance in the field of nucleic acid detection. Further, the method should be tested in more real samples.

I feel that this work does not reach the high threshold to make the manuscript a strong candidate for nBME. I would like to suggest that publication would be more appropriate in other journals about synthetic biology or analytical chemistry.

Reviewer #2 (Report for the authors (Required)):

The manuscript of nBME-22-1597 described a new RNA detection platform capable of multiplexing. Specificity of the reaction was achieved by ligation after hybridization, and reporter proteins translated by cell-free expression components provide the flexibility of output signals. The whole reaction could be run at ambient temperature (22°C) and does not require any device. Various pathogens can be detected individually in a single reaction, which makes it a potentially powerful tool in a point-of-care setting. Since this technology works not only on RNA targets but also on denatured DNA targets, it is expected to be useful in all fields requiring nucleic acid detection.

However, the sensitivity of the INSPECTR does not appear to be sufficient, and the reaction requires a long-expression time or 2 h of amplification step. It is clear that the INSPECTR has some properties suitable for field use, but more data/evidence/development seems required to make this technology useful in practical use.

Major comments:

1) Is it possible that the primer binding to the ssDNA probe B and elongation in advance of hybridization of probe A and B by RNA target? If it is possible, this reviewer thinks it may prevent RNA target binding to ssDNA probe B, which results in lowering the sensitivity of the test.

2) The reaction is primarily composed of two steps, ligation and cell-free expression, which are conducted in different reaction tubes (two-pot reaction). Therefore, it mitigates the advantages of the detection method. In fact, including pretreatment and LFA, the total reaction steps are four. Would it be possible to reduce the number of steps?

3) The strip order of antibodies seems to be different between figure 4 and figure 5. According to the article, the test strips' order was optimized considering the specificity of the reporter. How exactly was the order optimized? Is it possible to provide the related data in more detail?

4) anti-Myc (T1), anti-OLLAS (T2), and anti-HA (T7), from the test strip which was presented in Figure 4b, were excluded from the test strip of Figure 5. Regarding this, the authors said the test signal of T1 and T2 was low (Line 263-265), and probably the data is not shown in the manuscript. However, anti-Myc (T1) and anti-OLLAS (T2) showed higher intensity than anti-VSV-G (T3) in Figure 4b. Given this, should the detection epitope-ab pair be tested and optimized when the target sequence changes? Would this feature make the INSPECTR technology less modular to respond to the emerging infectious disease?

5) This reviewer would like to ask if there is any probe design rule to share. For example, more details about target regions (like G:C ratio or length or sequence-specificity) in which RNA: DNA hybridization occurs (Figure 1). Also, more information about gap-filling short 10-nt nucleotide for RCA reaction (Figure 5).

6) It would be better to include clinical trials with this INSPECTR technology. Especially, there are many SARS-CoV-2 or influenza A or B resources available to the research institution. This reviewer thinks including clinical trials will confirm the INSPECTR's performance and capability as a field-deployable diagnostics assay.

7) Overall reaction time is more than 2 h. For use as a point-of care diagnostic, it is thought that reaction time should be reduced.

8) Since the RCA used as pre-amplification (up to 2 h), there seem to be many limitations as a molecular diagnostic technology. How much does the LoD decrease if the reaction is performed with shorter than 2 h? Is there any data showing the change in LoD over time?

Minor comments:

9) Line 458, '5 μ L μ L' to '5 μ L'

10) Line 462 states that the ligation reaction takes 45 minutes, while line 176, below Figure 3, states it lasted for 30 minutes. Please make this clear.

11) For figure 4B, the plotted data of co-expressed assay (all) could not be distinguished clearly, particularly

the dot corresponding to anti-DEP5 (T3).

Reviewer #3 (Report for the authors (Required)):

Multiplexed detection of pathogen nucleic acids in the point-of-care setting is an important unmet need. Phillips et al present a novel multiplexed probe ligation based assay for detection of pathogen associated nucleic acids, using detection of SARS-CoV-2 genomic RNA as the timely use case. The authors appropriately explore different readout modalities, which illuminate their relative strengths and weaknesses. It is considered from the perspective of synthetic biology that this technology may one day be used to encode programmable therapeutics that are translated on demand in response to detection of the pathogen they're meant to treat. This is a compelling narrative that extends the rationale for the further development of this technology. The authors should be applauded for their strong work. My biggest issue was that the hands-on workflows tend to be a little opaque, forcing the reader to study the methods section to really gain an appreciation for how close or far this approach is from potentially being implemented in a diagnostic setting.

Minor comments:

- * The intro should include references to other probe ligation assays that have previously been used for RNA detection in the past, including RASL (in particular cRASL for pathogen RNA detection), temp-O-seq, etc.
- * Please make sure key abbreviations in the figures are defined in the legends (eg. "LOR" in Fig 3b).
- * The X axis of Fig 3B is hard to understand.
- * The nomenclature of the dual epitope tag in Fig 4 ("E1" and "E2") might be less confusing if the different E1s are made distinct in some way, for example "E1-i" or "E1a" etc.
- * Are the sequences or names of the 7 epitope tags provided somewhere?
- * The concentrations of synthetic RNA targets that generated Fig 5 were not clear.
- * On page 7, the 5-plex assay is referred to as "high-plex". This is a bit of an exaggeration. "multiplex" would be more appropriate.
- * It should be noted that the lateral flow strip was deposited into the translation reaction, not used in the typical format of a lateral flow test (wicking) to avoid misrepresenting the experimental setup.
- * "As shown in Fig. 6b, this amplification step improved the LoD of the assay to 3.5 fM of target RNA, using a fully ambient temperature workflow." Please report target copy number wherever discussing LoD or sensitivity throughout the manuscript.
- * While the GFO is expected to enhance specificity, it is also likely to reduce sensitivity since 2 ligations are required versus 1. Do the authors have any data on this loss of efficiency? If not, can it be estimated?
- * In the discussion, it is mentioned that cell extracts are "ultra-low-cost biochemical reagents". This is an exaggeration since commercially available products are anything but.
- * Please provide a more thorough discussion of the sensitivity of INSPECTR by comparing its sensitivity to other diagnostic assays.
- * In the methods, it is mentioned that probes B is used at 10x higher concentration vs probe A. Perhaps this could be noted and discussed in the results or discussion?
- * It would be very helpful to have a supplementary figure with more detailed hands-on workflows. What went into a tube with the sample, what happened to the tube, what was then added to the tube or taken out of the tube and added to another tube, etc, etc for each of the readouts described. A summary table of the pros and cons of each method explored might also be valuable for the reader.

Tue 13 Dec 2022

Decision on Article NBME-22-1597A

Dear Dr Joneja,

Thank you for your revised manuscript, "Multiplexed viral RNA detection with target-splinted ligation probes forming cell-free expression cassettes". Having consulted with the original reviewers (whose comments you will find at the end of this message), I am pleased to write that we shall be happy to publish the manuscript in *Nature Biomedical Engineering*.

We will be performing detailed checks on your manuscript, and in due course will send you a checklist detailing our editorial and formatting requirements. You will need to follow these instructions before you upload the final manuscript files.

Best wishes,

Pep

Pep Pàmies
Chief Editor, Nature Biomedical Engineering

Reviewer #1 (Report for the authors (Required)):

In this version, the authors have addressed all reviewers' concerns, and publication is recommended.

Reviewer #2 (Report for the authors (Required)):

The authors adequately answered the comments and questions raised by this reviewer and updated the manuscript accordingly. They now tested their method in a clinical sample matrix (saliva) and improved the sensitivity. Notably, as the authors also denoted in the text, the overall sensitivity falls between typical antigen assay and standard qPCR test. This reviewer believes this is a decent improvement, and it would have been possible because the authors introduced cell-free protein expression as a reporter system. In addition, they investigated the impact on sensitivity for each key step (ligation, amplification, and expression), added a step-by-step protocol, and clarified how they designed probe sets. This reviewer does not have any further comments and questions.

Reviewer #3 (Report for the authors (Required)):

The authors have adequately addressed my critiques.

Rebuttal 1

We thank the Reviewers for their time, thoughtful comments, and suggestions for improvement that helped us to strengthen our paper and to better present our findings. Below is our response to each point raised by the Reviewers. We hope that these responses and the revisions to the text (indicated in red in the revised manuscript file) adequately address the concerns that were noted, and that the manuscript is now suitable for publication in Nature Biomedical Engineering.

Reviewer #1 (Report for the authors (Required)):

The manuscript by Dr Joneja and colleagues presents an instrument-free INSPECTR method for multiplexed viral RNA detection based on target-splinted ligation and cell-free expression reactions. This method is very simple and practical, relies on the target-splinted ligation of DNA probes to make cell-free expression cassettes programmed for the synthesis of reporter proteins. In this work, small 45-amino acid alpha-fragment of LacZ, nanoluciferase (nLuc), and dual epitope peptides were investigated as reporter proteins for detection of five respiratory viral targets. Following a two-hour pre-amplification of RCA, the INSPECTR method enabled the detection of 3.5 fM of SARS-CoV-2 genomic RNA (4000 RNA copies) with a lateral flow readout. The data of the manuscript are comprehensive and convincing. The work is interesting and the presentation is adequate.

- We thank the Reviewer for their concise and accurate summary of our manuscript and are glad that they found it interesting and well-presented.

However, I think that its conceptual and methodological novelty is insufficient. The methods for detect of nucleic acids based on target-induced proximity ligation followed by amplification reaction or cell-free expression reactions have been well-established in past decades. The INSPECTR method did not provide a sufficiently striking advance and potential importance in the field of nucleic acid detection. Further, the method should be tested in more real samples. I feel that this work does not reach the high threshold to make the manuscript a strong candidate for nBME. I would like to suggest that publication would be more appropriate in other journals about synthetic biology or analytical chemistry.

- We acknowledge the Reviewer's criticism against the novelty of our method but respectfully disagree. To our knowledge, no published work has combined RNA-splinted ligation with cell-free expression, and we disagree that this is a "well-established" idea. We also do not believe that the presented work is an obvious follow-on to previous proximity ligation or cell-free expression technology. Only recently has splint-ligation even been combined with transcription of a fluorescence-activating aptamer, and this was considered sufficiently novel for publication in Nature BME (Woo, et. al. 2020). Our workflow is considerably distinct from that work, and more versatile, by including translation of reporter enzymes and peptides to enable non-instrumented detection. To that end, we also stress the novelty of our scheme for completely instrument-free, multiplexed, economical, and ambient-temperature detection of nucleic acids, a challenging aim that has not been demonstrated elsewhere. We additionally offer Reviewers' 2 and 3 positive perspectives on the manuscript's novelty as a counterpoint.

- We thank the Reviewer for the suggestion that the method should be tested on real clinical samples, and we agree that this would further improve the impact of the work. Using the nanoluciferase INSPECTR probe, we demonstrated robust detection of SARS-CoV-2 viral material in a clinical sample matrix (saliva), with 100% sensitivity and specificity at our indicated 10 fM (6,000 cps/ μ L) threshold in the sample; we also demonstrated detection of lower viral loads in a select clinical samples. These data are now presented in **Fig. S5** and referenced in the text.

Reviewer #2 (Report for the authors (Required)):

The manuscript of nBME-22-1597 described a new RNA detection platform capable of multiplexing. Specificity of the reaction was achieved by ligation after hybridization, and reporter proteins translated by cell-free expression components provide the flexibility of output signals. The whole reaction could be run at ambient temperature (22°C) and does not require any device. Various pathogens can be detected individually in a single reaction, which makes it a potentially powerful tool in a point-of-care setting. Since this technology works not only on RNA targets but also on denatured DNA targets, it is expected to be useful in all fields requiring nucleic acid detection.

- We thank the Reviewer for their concise and accurate summary of our manuscript, and statement on the technology's broad utility for nucleic acid detection.

However, the sensitivity of the INSPECTR does not appear to be sufficient, and the reaction requires a long-expression time or 2 h of amplification step. It is clear that the INSPECTR has some properties suitable for field use, but more data/evidence/development seems required to make this technology useful in practical use.

Major comments:

1) Is it possible that the primer binding to the ssDNA probe B and elongation in advance of hybridization of probe A and B by RNA target? If it is possible, this reviewer thinks it may prevent RNA target binding to ssDNA probe B, which results in lowering the sensitivity of the test.

- We thank the Reviewer for pointing out this potential failure mode. Please see response to comment below, which addresses this concern and our attempts to condense the workflow.

2) The reaction is primarily composed of two steps, ligation and cell-free expression, which are conducted in different reaction tubes (two-pot reaction). Therefore, it mitigates the advantages of the detection method. In fact, including pretreatment and LFA, the total reaction steps are four. Would it be possible to reduce the number of steps?

- We thank the Reviewer for these helpful technical suggestions and agree that shortening the reaction workflow would be broadly useful. The outcome of our study is in the new **Fig. S6**.
- We decided to test both strategies (primer extension on the template before ligation and trying to one-pot individual reaction steps) using a new INSPECTR probe that encodes just the 11-amino acid HiBiT peptide, a complement of the split nanoluciferase (**Fig. S3a**). Since HiBiT is significantly shorter and simpler than the full-length nLuc protein, we hypothesized this probe would be robust to structural interference and could be a useful quantitative model system for optimizing the assay. As with nLuc, expression of HiBiT is linear with expression cassette at low fM concentrations of the expression cassette (**Fig. S3b**) and its LoD in a two-pot ligation-expression assay is around 1 pM, consistent with our previous estimates of SplintR ligation efficiency (**Fig. S3c**). We also used this assay in **Fig. S3d-e** to interrogate the mechanism of the gap-filling oligo, since the short probe shows undetectable amounts of ligase-independent "leak" or off-target ligation (see response below).
- Measured both by qPCR (using primers that span the ligation junction) and HiBiT expression (**Fig. S6**), we confirmed that the ligation must be run before primer extension and cell-free expression in a separate pot because occluding the ligation junction with double-stranded DNA blocks RNA

splinting. We suspect that the ligation is simply slower than double-stranding, but it necessarily must happen first. This is consistent with previous work for padlock probes, in which the RCA reaction is run in a separate pot from the ligation^{1,2}.

3) The strip order of antibodies seems to be different between figure 4 and figure 5. According to the article, the test strips' order was optimized considering the specificity of the reporter. How exactly was the order optimized? Is it possible to provide the related data in more detail?

- We thank the reviewer for their attention to detail; in fact, we made a labeling error and have amended the methods (see Line 526) to reflect that the antibodies in Figure 4 are striped as anti-Ty1 (T7), anti-HA (T6), anti-VSV-G (T5), anti-S-tag (T4), anti-Myc (T3), anti-V5 (T2), and anti-OLLAS (T1). We have also added additional commentary on strategies for selecting the striping order (see Line 232).

4) anti-Myc (T1), anti-OLLAS (T2), and anti-HA (T7), from the test strip which was presented in Figure 4b, were excluded from the test strip of Figure 5. Regarding this, the authors said the test signal of T1 and T2 was low (Line 263-265), and probably the data is not shown in the manuscript. However, anti-Myc (T1) and anti-OLLAS (T2) showed higher intensity than anti-VSV-G (T3) in Figure 4b. Given this, should the detection epitope-ab pair be tested and optimized when the target sequence changes? Would this feature make the INSPECTR technology less modular to respond to the emerging infectious disease?

- We thank the reviewer again for their attention to detail; with the amended labeling of the striping order, we believe that the reviewer will agree that the anti-Myc (T3) and anti-OLLAS (T1) test lines yielded the faintest test lines and so it was appropriate to remove them from further development. We also added the FLAG tag into the test strip of Figure 5 (and removed the HA tag) because we had substantially more data associated with that epitope in combination with SARS-CoV2-WT detection (as can be seen in the data associated with the universal test strip in Figure 4A). With this information and clarification, we hope the reviewer would agree that the modularity of the peptide readout provides an advantage for detection of varying target sequences.

5) This reviewer would like to ask if there is any probe design rule to share. For example, more details about target regions (like G:C ratio or length or sequence-specificity) in which RNA: DNA hybridization occurs (Figure 1). Also, more information about gap-filling short 10-nt nucleotide for RCA reaction (Figure 5).

- We thank the Reviewer for their interest in the technology and design rules. We provide additional data supporting the use of the GFO in the new **Fig. S3**. For the design of the target regions, we aimed to maximize conservation in viral genomes above any sequence preferences, noting previous studies that indicate SplintR ligates RNA well as long as the overhangs are above 4-6 nt long.^{3,4} We kept the junction roughly symmetric and tried to ensure that the 5' phosphate donor nucleotide was not dG or dC, in line with the enzyme's sequence preference.³ We have added this text to the body of the manuscript (see Line 150).

6) It would be better to include clinical trials with this INSPECTR technology. Especially, there are many SARS-CoV-2 or influenza A or B resources available to the research institution. This reviewer

thinks including clinical trials will confirm the INSPECTR's performance and capability as a field-deployable diagnostics assay.

- We thank the Reviewer for the suggestion that the method should be tested on real clinical samples and we agree that this would further improve the impact of the work. Using the nanoluciferase INSPECTR probe, we demonstrated robust detection of SARS-CoV-2 viral material in a clinical sample matrix (saliva), with 100% sensitivity and specificity at our indicated 10 fM (6,000 cps/ μ L) threshold in the sample; we were also able to detect lower viral loads in a few samples as well. These data are now presented in **Fig. S5** and indicated in the text (see Line 369).

7) Overall reaction time is more than 2 h. For use as a point-of care diagnostic, it is thought that reaction time should be reduced.

- The authors agree that a shorter turnaround time would be valuable for a point-of-care diagnostic and acknowledge that the methods presented in this manuscript do not describe a mature solution for any particular diagnostic assay. The authors also believe that for point-of-need applications, where an end user may be running the assay in their home, there is more flexibility for turnaround time as compared to point-of-care scenarios where a result is most beneficial prior to the patient leaving the healthcare institution. We have modified some text to clarify the utility of INSPECTR at the point-of-need (see Line 394).

8) Since the RCA used as pre-amplification (up to 2 h), there seem to be many limitations as a molecular diagnostic technology. How much does the LoD decrease if the reaction is performed with shorter than 2 h? Is there any data showing the change in LoD over time?

- We thank the Reviewer for this helpful suggestion, and we decided to individually test the impact on sensitivity for each of the key “long” steps—ligation, amplification, and expression. The summary of these data are now in **Fig. S7**.
- Briefly, attempts to reduce the RCA time or the expression time result in decreased sensitivity. We have also observed that extending these reaction times beyond 2 hours have diminishing returns and do not significantly improve sensitivity. We did find that increasing the ligation time was effective at improving the signal from a luminescent HiBiT probe (**Fig. S3**), perhaps as expected by the low catalytic rate of SplintR ligase, but we did not pursue this as a way of improving the LoD since amplification of the ligated probe is a more efficient use of assay time.

Minor comments:

9) Line 458, '5 μ L μ L' to '5 μ L'

- We thank the reviewer for catching this typo and have corrected it in the text (now Line 482).

10) Line 462 states that the ligation reaction takes 45 minutes, while line 176, below Figure 3, states it lasted for 30 minutes. Please make this clear.

- We thank the reviewer for pointing out this discrepancy; the correct time is 30 minutes, and this has been updated in the text (now Line 489).

11) For figure 4B, the plotted data of co-expressed assay (all) could not be distinguished clearly, particularly the dot corresponding to anti-DEP5 (T3).

- We thank the reviewer for pointing this out and have amended the plot to have a break in the axis and smaller data points, which should make it easier to identify the individual expression conditions.

Reviewer #3 (Report for the authors (Required)):

Multiplexed detection of pathogen nucleic acids in the point-of-care setting is an important unmet need. Phillips et al present a novel multiplexed probe ligation based assay for detection of pathogen associated nucleic acids, using detection of SARS-CoV-2 genomic RNA as the timely use case. The authors appropriately explore different readout modalities, which illuminate their relative strengths and weaknesses. It is considered from the perspective of synthetic biology that this technology may one day be used to encode programmable therapeutics that are translated on demand in response to detection of the pathogen they're meant to treat. This is a compelling narrative that extends the rationale for the further development of this technology. The authors should be applauded for their strong work. My biggest issue was that the hands-on workflows tend to be a little opaque, forcing the reader to study the methods section to really gain an appreciation for how close or far this approach is from potentially being implemented in a diagnostic setting.

- We thank the Reviewer for their overall positive perspective on our manuscript. We broadly agree that the methods and workflows are complex and have therefore provided a Supplemental File providing a detailed step-by-step protocol of the INSPECTR workflow and process. We hope that this information will facilitate others' adoption and optimization of this technology.

Minor comments:*

The intro should include references to other probe ligation assays that have previously been used for RNA detection in the past, including RASL (in particular cRASL for pathogen RNA detection), temp-O-seq, etc.

- We thank the Reviewer for pointing out this omission and now include a number of citations to previous work in this vein (see Line 72).

*** Please make sure key abbreviations in the figures are defined in the legends (eg. "LOR" in Fig 3b).**

- We thank the Reviewer for this suggestion and have reviewed the manuscript to confirm that all abbreviations are clearly defined.

*** The X axis of Fig 3B is hard to understand.**

- We thank the Reviewer for pointing out this source of confusion and have amended the figure to make it clearer.

*** The nomenclature of the dual epitope tag in Fig 4 ("E1" and "E2") might be less confusing if the different E1s are made distinct in some way, for example "E1-i" or "E1a" etc.**

- We thank the Reviewer for this suggestion and agree that it would make it easier to interpret. We have amended the figure as suggested.

*** Are the sequences or names of the 7 epitope tags provided somewhere?**

- Yes, the names of the epitopes are listed in the methods section "lateral flow strip preparation" and the oligonucleotide sequences of the expression cassettes used to generate the epitope tags are listed in the Supplementary Table 1.

*** The concentrations of synthetic RNA targets that generated Fig 5 were not clear.**

- 10 nM of synthetic RNA target was used to generate Figure 5, as stated in the methods section “Dual epitope peptide expression and detection.” We have added “Fig. 4b” and “Fig. 5” into the methods section to help future readers find the relevant methods for each figure.

*** On page 7, the 5-plex assay is referred to as “high-plex”. This is a bit of an exaggeration. “multiplex” would be more appropriate.**

- We thank the Reviewer for this suggestion and have amended the text to state that we “successfully enabled five-plex detection” (see Line 282).

*** It should be noted that the lateral flow strip was deposited into the translation reaction, not used in the typical format of a lateral flow test (wicking) to avoid misrepresenting the experimental setup.**

- We thank the Reviewer for suggesting this clarification. The “dipstick” format is now explicitly stated and graphically represented in the step-by-step protocol that has been appended to the Supplemental Information.

*** “As shown in Fig. 6b, this amplification step improved the LoD of the assay to 3.5 fM of target RNA, using a fully ambient temperature workflow.” Please report target copy number wherever discussing LoD or sensitivity throughout the manuscript.**

- We thank the Reviewer for pointing out this omission, and the target copy number has been added to the text wherever sensitivity is discussed (for that example, see Line 328).

*** While the GFO is expected to enhance specificity, it is also likely to reduce sensitivity since 2 ligations are required versus 1. Do the authors have any data on this loss of efficiency? If not, can it be estimated?**

- We thank the Reviewer for pointing this out and agreed that it would be a useful quantitative comparison to generate. The results of this study are now in **Fig. S3**.
- We decided to test the hypothesis using new INSPECTR probes that encode just the 11-amino acid HiBiT peptide, a complement of the split nanoluciferase (**Fig. S3 a-c**). Since HiBiT is significantly shorter and simpler than the full-length nLuc protein, we hypothesized that this probe would also be robust to structural interference or self-splinting. This would allow us to more easily compare ON ligation efficiency without detecting off-target signal.
- We constructed probes in which the HiBiT sequence was split into either two or three segments. Measuring by both qPCR and HiBiT expression, we found that, as the Reviewer hypothesizes, additional ligation junctions do reduce the efficiency, but only by approximately 1-2 Cts (qPCR) or a protein expression yield difference of 3-4X (**Fig. S3d**). The difference in specificity appears to be more important for probes that show significant off-target-ligation-dependent background (e.g., the RCA-amplified probes in **Fig. 6**). The study and reduction in ligation efficiency is summarized in Line 270.

*** In the discussion, it is mentioned that cell extracts are “ultra-low-cost biochemical reagents”. This is an exaggeration since commercially available products are anything but.**

- We thank the Reviewer for their perspective and acknowledge that the commercial cell-free expression kits are expensive. However, in **Fig. 2**, we show that homemade extract performs at least as well as commercial extract and, in some cases, superior to the reconstituted PURE Express system as well. Significant recent work has shown that costs can be significantly brought down using, e.g., sugars as the energy source instead of PEP for energy regeneration.⁵⁻⁹
- To ensure broader reproducibility between experiments, we did perform most of our INSPECTR experiments using commercial systems (PURE Express or NEB Express). However, the workflow is compatible with our in-house extracts: indeed, the LacZ data in **Fig. S4** had to be run with an in-house prepared extract from a LacZ-deficient strain, since commercial extracts have very high background beta-galactosidase activity.

*** Please provide a more thorough discussion of the sensitivity of INSPECTR by comparing its sensitivity to other diagnostic assays.**

- We thank the Reviewer for their suggestion to provide context to the sensitivity of INSPECTR. The reported analytical sensitivity of lateral flow tests and PCR-based assays varies widely, but it can be stated that INSPECTR occupies the very large middle ground between the two technologies. This statement and reviews with supporting data have been added to the Discussion section of the manuscript (see Line 380).

*** In the methods, it is mentioned that probes B is used at 10x higher concentration vs probe A. Perhaps this could be noted and discussed in the results or discussion?**

- We thank the Reviewer for this suggestion and have noted in the methods section that the linear probe concentrations were chosen to maximize signal and reduce leak; increasing probe B but not probe A to 3.8 pM led to higher leak, so we maximized probe concentration where possible. This is noted on Line 485.

*** It would be very helpful to have a supplementary figure with more detailed hands-on workflows. What went into a tube with the sample, what happened to the tube, what was then added to the tube or taken out of the tube and added to another tube, etc, etc for each of the readouts described.**

- We thank the Reviewer for this suggestion. A detailed step-by-step protocol describing the volumes, timing, user steps, and reagents associated with each reaction has been appended to the Supplemental Information. If the manuscript is accepted, this protocol will also be uploaded to the Protocol Exchange to make it freely accessible.

A summary table of the pros and cons of each method explored might also be valuable for the reader.

- We thank the Reviewer for the suggestion of a summary table, and have included Supplementary Table S2 to provide a quick reference of the pros and cons for each explored reporter protein. This table is referenced in the Discussion section (see Line 400).

1. Lizardi, P. M.; Huang, X.; Zhu, Z.; Bray-Ward, P.; Thomas, D. C.; Ward, D. C., Mutation detection and single-molecule counting using isothermal rolling-circle amplification. *Nature Genetics* **1998**, *19* (3), 225-232.
2. Mignardi, M.; Mezger, A.; Qian, X.; La Fleur, L.; Botling, J.; Larsson, C.; Nilsson, M., Oligonucleotide gap-fill ligation for mutation detection and sequencing in situ. *Nucleic Acids Res* **2015**, *43* (22), e151.
3. Lohman, G. J.; Zhang, Y.; Zhelkovsky, A. M.; Cantor, E. J.; Evans, T. C., Jr., Efficient DNA ligation in DNA-RNA hybrid helices by Chlorella virus DNA ligase. *Nucleic Acids Res* **2014**, *42* (3), 1831-44.
4. Krzywkowski, T.; Nilsson, M., Fidelity of RNA templated end-joining by chlorella virus DNA ligase and a novel iLock assay with improved direct RNA detection accuracy. *Nucleic Acids Research* **2017**, *45* (18), e161-e161.
5. Au - Grasmann, L.; Au - Lavickova, B.; Au - Elizondo-Cantú, M. C.; Au - Maerkl, S. J., OnePot PURE Cell-Free System. *JoVE* **2021**, (172), e62625.
6. Levine, M. Z.; Gregorio, N. E.; Jewett, M. C.; Watts, K. R.; Oza, J. P., Escherichia coli-Based Cell-Free Protein Synthesis: Protocols for a robust, flexible, and accessible platform technology. *JoVE* **2019**, (144), e58882.
7. Warfel, K. F.; Williams, A.; Wong, D. A.; Sobol, S. E.; Desai, P.; Li, J.; Chang, Y.-F.; DeLisa, M. P.; Karim, A. S.; Jewett, M. C., A low-cost, thermostable, cell-free protein synthesis platform for on demand production of conjugate vaccines. *bioRxiv* **2022**, 2022.08.10.503507.
8. Guzman-Chavez, F.; Arce, A.; Adhikari, A.; Vadhin, S.; Pedroza-Garcia, J. A.; Gandini, C.; Ajioka, J. W.; Molloy, J.; Sanchez-Nieto, S.; Varner, J. D.; Federici, F.; Haseloff, J., Constructing Cell-Free Expression Systems for Low-Cost Access. *ACS Synthetic Biology* **2022**, *11* (3), 1114-1128.
9. Arce, A.; Guzman Chavez, F.; Gandini, C.; Puig, J.; Matute, T.; Haseloff, J.; Dalchau, N.; Molloy, J.; Pardee, K.; Federici, F., Decentralizing Cell-Free RNA Sensing With the Use of Low-Cost Cell Extracts. *Frontiers in Bioengineering and Biotechnology* **2021**, *9*.